# Investigations on Dynamical Stability in 3D Quadrupole Ion Traps

**Bogdan M. Mihalcea** [1,*] and **Stephen Lynch** [2]

[1] National Institute for Laser, Plasma and Radiation Physics (INFLPR), Atomiştilor Str. Nr. 409, 077125 Măgurele, Romania
[2] Department of Computing and Mathematics, Manchester Metropolitan University, Manchester M1 5GD, UK; s.lynch@mmu.ac.uk
\* Correspondence: bogdan.mihalcea@inflpr.ro; Tel.: +40-21-4574437

**Abstract:** We firstly discuss classical stability for a dynamical system of two ions levitated in a 3D Radio-Frequency (RF) trap, assimilated with two coupled oscillators. We obtain the solutions of the coupled system of equations that characterizes the associated dynamics. In addition, we supply the modes of oscillation and demonstrate the weak coupling condition is inappropriate in practice, while for collective modes of motion (and strong coupling) only a peak of the mass can be detected. Phase portraits and power spectra are employed to illustrate how the trajectory executes quasiperiodic motion on the surface of torus, namely a Kolmogorov–Arnold–Moser (KAM) torus. In an attempt to better describe dynamical stability of the system, we introduce a model that characterizes dynamical stability and the critical points based on the Hessian matrix approach. The model is then applied to investigate quantum dynamics for many-body systems consisting of identical ions, levitated in 2D and 3D ion traps. Finally, the same model is applied to the case of a combined 3D Quadrupole Ion Trap (QIT) with axial symmetry, for which we obtain the associated Hamilton function. The ion distribution can be described by means of numerical modeling, based on the Hamilton function we assign to the system. The approach we introduce is effective to infer the parameters of distinct types of traps by applying a unitary and coherent method, and especially for identifying equilibrium configurations, of large interest for ion crystals or quantum logic.

**Keywords:** radiofrequency trap; dynamical stability; eigenfrequency; Paul and Penning trap; Hessian matrix; Hamilton function; bifurcation diagram

**PACS:** 37.10.Ty; 02.30.-f; 02.40.Xx

## 1. Introduction

The advent of ion traps has led to remarkable progress in modern quantum physics, in atomic and nuclear physics or quantum electrodynamics (QED) theory. Experiments with ion traps enable ultrahigh resolution spectroscopy experiments, quantum metrology measurements of fundamental quantities such as the electron and positron $g$-factors [1], high precision measurements of the magnetic moments of leptons and baryons (elementary particles) [2], Quantum Information Processing (QIP) and quantum metrology [3,4], etc.

Scientists can now trap single atoms or photons, acquire excellent control on their quantum states (inner and outer degrees of freedom) and precisely track their evolution by the time [5]. A single ion or an ensemble of ions can be secluded with respect to external perturbations, then engineered in a distinct quantum state and trapped in ultrahigh vacuum for a long period of time (operation times of months to years) [6–8], under conditions of dynamical stability. Under these circumstances, the superposition states required for quantum computation can live for a relatively long time. Ion localization results in unique features such as extremely high atomic line quality factors under minimum perturbation by the environment [3]. The remarkable control accomplished by employing ion traps and

laser cooling techniques has resulted in exceptional progress in quantum engineering of space and time [2,9]. Nevertheless, trapped ions can not be completely decoupled from the interaction with the surrounding environment, which is why new elaborate interrogation and detection techniques are continuously developed and refined [6].

These investigations allow scientists to perform fundamental tests of quantum mechanics and general relativity, to carry out matter and anti-matter tests of the Standard Model, to achieve studies with respect to the spatio-temporal variation of the fundamental constants in physics at the cosmological scale [2], or to perform searches for dark matter, dark energy and extra forces [9]. Moreover, there is a large interest towards quantum many-body physics in trap arrays with an aim to achieve systems of many interacting spins, represented by qubits in individual microtraps [6,10].

The trapping potential in a Radio-Frequency (RF) trap harmonically confines ions in the region where the field exhibits a minimum, under conditions of dynamical stability [1,11,12]. Hence, a trapped ion can be regarded as a quantum harmonic oscillator [13–15].

A problem of large interest concerns the strong outcome of the trapped ion dynamics on the achievable resolution of many experiments, and the paper builds exactly in this direction. Fundamental understanding of this problem can be achieved by using analytical and numerical methods which take into account different trap geometries and various cloud sizes. The other issue lies in performing quantum engineering (quantum optics) experiments and high-resolution measurements by developing and implementing different interaction protocols.

*Investigations on Classical and Quantum Dynamics Using Ion Traps*

A detailed experimental and theoretical investigation with respect to the dynamics of two-, three-, and four-ion crystals close to the Mathieu instability is presented in [16], where an analytical model is introduced that is later used in a large number of papers to characterize regular and nonlinear dynamics for systems of trapped ions in 3D QIT. We use this model in our paper and extend it. Numerical evidence of quantum manifestation of order and chaos for ions levitated in a Paul trap is explored in [17], where it is suggested that at the quantum level one can use the quasienergy states statistics to discriminate between integrable and chaotic regimes of motion. Double well dynamics for two trapped ions (in a Paul or Penning trap) is explored in [18], where the RF-drive influence in enhancing or modifying quantum transport in the chaotic separatrix layer is also discussed. Irregular dynamics of a single ion confined in electrodynamic traps that exhibit axial symmetry is explored in [19], by means of analytical and numerical methods. It is also established that period-doubling bifurcations represent the preferred route to chaos.

Ion dynamics of a parametric oscillator in a RF octupole trap is examined in [20,21] with particular emphasis on the trapping stability, which is demonstrated to be position dependent. In Ref. [22] quantum models are introduced to describe multi body dynamics for strongly coupled Coulomb systems SCCS [23] stored in a 3D QIT that exhibits axial (cylindrical) symmetry.

A trapped and laser cooled ion that undergoes interaction with a succession of stationary-wave laser pulses may be regarded as the realization of a parametric nonlinear oscillator [24]. Ref. [25] uses numerical methods to explore chaotic dynamics of a particle in a nonlinear 3D QIT trap, which undergoes interaction with a laser field in a quartic potential, in presence of an anharmonic trap potential. The equation of motion is similar to the one that portrays a forced Duffing oscillator with a periodic kicking term. Fractal attractors are identified for special solutions of ion dynamics. Similarly to Ref. [19], frequency doubling is demonstrated to represent the favourite route to chaos. An experimental confirmation of the validity of the results obtained in [25] can be found in Ref. [26], where stable dynamics of a single trapped and laser cooled ion oscillator in the nonlinear regime is explored.

Charged microparticles stored in a RF trap are characterized either by periodic or irregular dynamics, where in the latter case chaotic orbits occur [27]. Ref. [28] explores dynamical stability for an ion confined in asymmetrical, planar RF ion traps, and establishes that the equations of motion are coupled. Quantum dynamics of ion crystals in RF traps is explored in [29] where stable trapping is discussed along with the validity of the pseudopotential approximation. A phase space study of surface electrode ion traps (SET) that explores integrable, chaotic and combined dynamics is performed in [30], with an emphasis on the integrable and chaotic motion of a single ion. The nonlinear dynamics of an electrically charged particle stored in a RF multipole ion trap is investigated in [31]. An in-depth study of the random dynamics of a single trapped and laser cooled ion that emphasizes nonequilibrium dynamics, is performed in Ref. [32]. Classical dynamics and dynamical quantum states of an ion are investigated in [33], considering the effects of the higher order terms of the trap potential. On the other hand, the method suggested in [34] can be employed to characterize ion dynamics in 2D and 3D QIT traps.

All these experimental and analytical investigations previously described open new directions of action towards an in-depth exploration of the dynamical equilibrium at the atomic scale, as the subject is extremely pertinent. In our paper we perform a classical study of the dynamical stability for trapped ion systems in Section 2, based on the model introduced in [16,18]. The associated dynamics is shown to be quasiperiodic or periodic. We use the dynamical systems theory to characterize the time evolution of two coupled oscillators in a RF trap, depending on the chosen control parameters. We consider the pseudopotential approximation, where the motion is integrable only for discrete values of the ratio between the axial and the radial frequencies of the secular motion. In Section 3 we apply the Morse theory to qualitatively analyze system stability. The results are extended to many body strongly coupled trapped ion systems, locally studied in the vicinity of equilibrium configurations that identify ordered structures. These equilibrium configurations exhibit a large interest for ion crystals or quantum logic. Section 4 explores quantum stability and ordered structures for many body dynamics (assuming the ions are identical) in a RF trap. We find the system energy and introduce a method (model) that supplies the elements of the Hessian matrix of the potential function for a critical point. Section 5 applies the model suggested in Section 4. Collective models are introduced and we build integrable Hamiltonians which admit dynamic symmetry groups. We particularize this Hamiltonian function for systems of trapped ions in combined (Paul and Penning) traps with axial symmetry. An improved model results by which multi-particle dynamics in a 3D QIT is associated with dynamic symmetry groups [35–37] and collective variables. The ion distribution in the trap can be described by employing numerical programming, based on the Hamilton function we obtain. This alternative technique can be very helpful to perform a unitary description of the parameters of different types of traps in an integrated approach. We emphasize the contributions in Section 6 and discuss the potential area of applications in Section 7.

## 2. Analytical Model

### 2.1. Dynamical Stability for Two Coupled Oscillators in a Radiofrequency Trap

We use the dynamical systems theory to investigate classical stability for two coupled oscillators (ions) of mass $m_1$ and $m_2$, respectively, levitated in a 3D radiofrequency RF QIT. The constants of force are denoted as $k_1$ and $k_2$, respectively. Ion dynamics restricted to the $xy$-plane is described by a set of coupled equations:

$$\begin{cases} m_1\ddot{x} = -k_1 x + b(x - y) \\ m_2\ddot{y} = -k_2 y - b(x - y) \, , \end{cases} \tag{1}$$

where $b$ stands for the parameter that characterizes Coulomb repulsion between ions. The control parameters for the trap are:

$$k_i = \frac{m_i q_i^2 \Omega}{8} \ , \quad q_i = 4 \frac{Q_i}{m_i} \frac{V_0}{(z_0 \Omega)^2} \ ; \ i = 1, 2, \tag{2}$$

with $\Omega$ the frequency of the micromotion, $V_0$ denotes the RF trapping voltage, $z_0$ is the trap axial dimension, $Q_i$ represents the electric charge and $m_i$ stands for the mass of the ion labeled as $i$. We use the time-independent (also known as pseudopotential) approximation of the RF trap electric potential, because it can be employed to achieve a good description of stochastic dynamics [32]. As heating of ion motion occurs in our case due to the Coulomb interaction between ions (as an outcome of energy transfer from the trapping field to the ions), the time-independent approximation can be safely used, which brings a significant simplification to the problem. Therefore, higher order terms in the Mathieu equation [1,38] that portrays ion motion can be discarded.

The simplest non-trivial model to describe the dynamic behavior is the Hamilton function of the relative motion of two levitated ions that interact via the Coulomb force in a 3D QIT that exhibits axial symmetry, under the time-independent approximation (autonomous Hamiltonian) [16–18]. The paper uses this well established model, which we extend.

We consider the electric potential to be a general solution of the Laplace equation, built using spherical harmonics functions with time dependent coefficients (see Appendix A). This family of potentials accounts for most of the ion traps that are used in experiments [29]. The Coulomb constant of force is $b \equiv 2Q^2/r^3 < 0$, resulting from a series expansion of $Q^2/r^2$ about a mean deviation of the ion with respect to the trap centre $r_0 \equiv (x_0 - y_0) < 0$, established by the initial conditions.

The expressions of the kinetic and potential energy are:

$$T = \frac{m_1 \dot{x}^2}{2} + \frac{m_2 \dot{y}^2}{2} \ , \quad U = \frac{k_1 x^2}{2} + \frac{k_2 y^2}{2} + \frac{1}{2} b(x - y)^2. \tag{3}$$

It is assumed that the ions share equal electric charges $Q_1 = Q_2$. We denote

$$k_1 = 2Q_1 \beta_1 \ , \quad \beta_1 = \frac{4Q_1 V_0^2}{m_1 \Omega^2 \xi^4} - \frac{2U_0}{\xi^2} \ , \tag{4}$$

with $\xi^2 = r_0^2 + 2z_0^2$, where $r_0$ and $z_0$ denote the radial and axial trap semiaxes. We assume $U_0 = 0$ (the d.c. trapping voltage) and consider $r_0$ as negligible. The trap control parameters are $U_0, V_0, \xi$ and $k_i$. We select an electric potential $V = 1/|z|$ and we perform a series expansion around $z_0 > 0$, with $z - z_0 = x - y$. The potential energy can be then cast into:

$$U = \frac{k_1 x_1^2}{2} + \frac{k_2 x_2^2}{2} + \frac{1}{4\pi \varepsilon_0} \frac{Q_1 Q_2}{|x_1 - x_2|} \ . \tag{5}$$

The Hamilton principle states the system is stable if the potential energy $U$ exhibits a minimum

$$k_{1,2} x_{1,2} \mp \lambda = 0 \ , \quad \lambda = \frac{1}{4\pi \varepsilon_0} \frac{Q_1 Q_2}{|x_1 - x_2|^2} \ . \tag{6}$$

Then

$$k_1 x_1 + k_2 x_2 = 0 \ , \ x_{1\,\text{min}} = \frac{\lambda}{k_1} \ , \ x_{2\,\text{min}} = -\frac{\lambda}{k_2} \ . \tag{7}$$

Equation

$$\lambda = \sqrt[3]{\frac{1}{4\pi \varepsilon_0} \frac{k_1^2 k_2^2}{(k_1 + k_2)^2} Q_1 Q_2} \ , \tag{8}$$

supplies the points of minimum, $x_1$ and $x_2$, for an equilibrium state. We choose

$$z_0 = x_{1\,\text{min}} - x_{2\,\text{min}} = \lambda \frac{k_1 k_2}{k_1 + k_2} \tag{9}$$

and denote

$$z = x_1 - x_2 , \ x_1 = x_{1\,\text{min}} + x , \ x_2 = x_{2\,\text{min}} + y , \tag{10}$$

with $z = z_0 + x - y$. Equation (8) gives us

$$\frac{Q_1 Q_2}{4\pi\varepsilon_0} = \lambda^3 \left( \frac{k_1 k_2}{k_1 + k_2} \right)^2 = \lambda z_0^2 . \tag{11}$$

We turn back to Equation (5), then make use of Equations (7) and (10) to express the potential energy as

$$U = \frac{k_1}{2} \left( x_{1\,\text{min}}^2 + x^2 \right) + \frac{k_2}{2} \left( x_{2\,\text{min}}^2 + y^2 \right) + \lambda z_0 + \frac{\lambda}{z_0} (z - z_0)^2 - \dots \tag{12}$$

From Equation (3) we obtain $b/2 = \lambda/z_0$. When the potential energy is minimum the system is stable.

## 2.2. Solutions of Coupled System of Equations

We seek for a stable solution of the coupled system of Equation (1) of the form

$$x = A \sin \omega t , \ y = B \sin \omega t . \tag{13}$$

The Wronskian determinant of the resulting system of equations must be zero for a stable system

$$\begin{vmatrix} b - k_1 + m_1 \omega^2 & -b \\ -b & b - k_2 + m_2 \omega^2 \end{vmatrix} = 0 . \tag{14}$$

The determinant allows us to construct the characteristic equation:

$$\left( b - k_1 + m_1 \omega^2 \right) \left( b - k_2 + m_2 \omega^2 \right) - b^2 = 0 . \tag{15}$$

The discriminant of Equation (15) can be cast as

$$\Delta = [m_1 (b - k_2) - m_2 (b - k_1)]^2 + 4 m_1 m_2 b^2 . \tag{16}$$

The system admits solutions if the determinant is zero, as stated above. Hence, a solution of Equation (15) would be:

$$\omega_{1,2}^2 = \frac{m_1 (k_2 - b) + m_2 (k_1 - b) \pm \sqrt{\Delta}}{2 m_1 m_2} . \tag{17}$$

Then, we find a stable solution for the system of coupled oscillators

$$\begin{aligned} x_1 &= C_1 \sin(\omega_1 t + \varphi_1) + C_2 \sin (\omega_2 t + \varphi_2) , \\ y_1 &= C_3 \sin(\omega_1 t + \varphi_3) + C_4 \sin (\omega_2 t + \varphi_4) , \end{aligned} \tag{18}$$

which describes a superposition of two oscillations characterized by the secular frequencies $\omega_1$ and $\omega_2$, that is to say the system eigenfrequencies. Assuming that $b \ll k_{1,2}$ in Equation (15), the strong coupling condition is

$$\left| \frac{b}{k_i} \right| \gg \left| \frac{m_1 - m_2}{m_2} \right| , \tag{19}$$

where the modes of oscillation are

$$\omega_1^2 = \frac{1}{2}\left(\frac{k_1}{m_1} + \frac{k_2}{m_2}\right), \tag{20}$$

$$\omega_2^2 = \frac{1}{2}\left(\frac{k_1 - 2b}{m_1} + \frac{k_2 - 2b}{m_2}\right). \tag{21}$$

By exploring the phase relations between the solutions of Equation (1), we can ascertain that the $\omega_1$ mode corresponds to a translation of the ions (the distance $r_0$ between ions does not fluctuate), while the Coulomb repulsion remains steady as $b$ is absent in Equation (20). The axial current produced by this mode of translation can be detected (electronically). In the $\omega_2$ mode the distance between the ions fluctuates about a fixed centre of mass (CM), case when both the electric current and signal are zero. Optical detection is possible in the $\omega_2$ mode [39] even if electronic detection is not feasible. As a consequence, for collective modes of motion only a peak of the mass is detected that corresponds to the ion average mass. In case of weak coupling the inequality in Equation (19) overturns, and from Equation (17) we derive

$$\omega_{1,2}^2 = (k_{1,2} - b)/m_{1,2}, \tag{22}$$

which means that each mode of the dynamics matches a single mass, while resonance is shifted with the parameter $b$. In addition, within the limit of equal ion mass $m_1 = m_2$, the strong coupling requirement in Equation (19) is always fulfilled regardless of how weak is the Coulomb coupling. This renders the weak coupling condition unsuitable in practice.

For the stable solution described by Equation (18), we supply below the phase portraits for the coupled oscillators system, as shown in Figures 1–8.

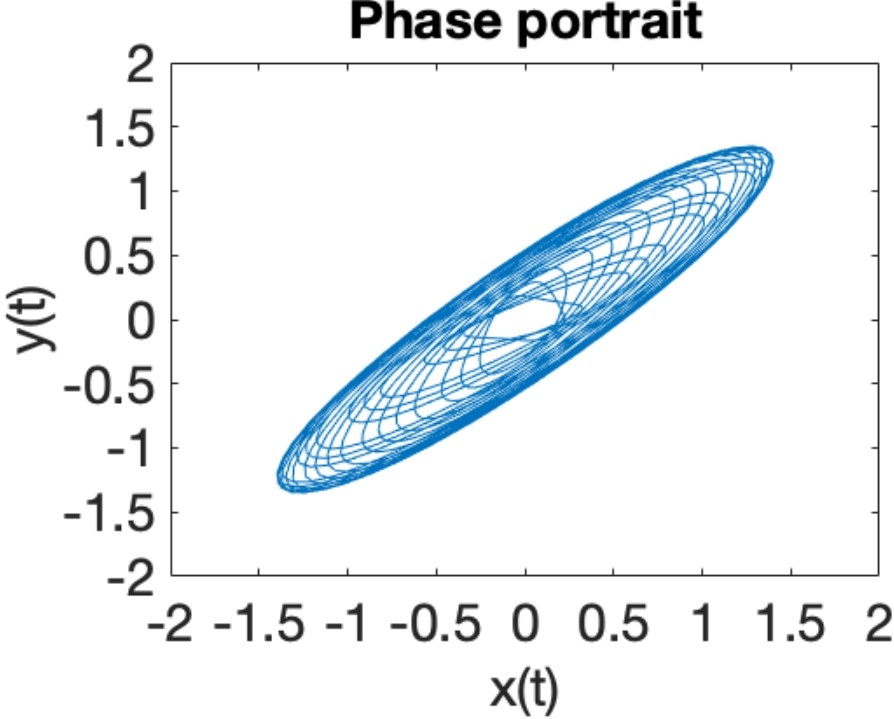

**Figure 1.** Parameter values $C_1 = 0.8, C_2 = 0.6, C_3 = 0.75, C_4 = 0.6, \varphi_1 = \pi/3, \varphi_2 = \pi/4, \varphi_3 = \pi/2, \varphi_4 = \pi/3, \omega_1/\omega_2 = 1.71/1.93 \longmapsto$ quasiperiodic dynamics.

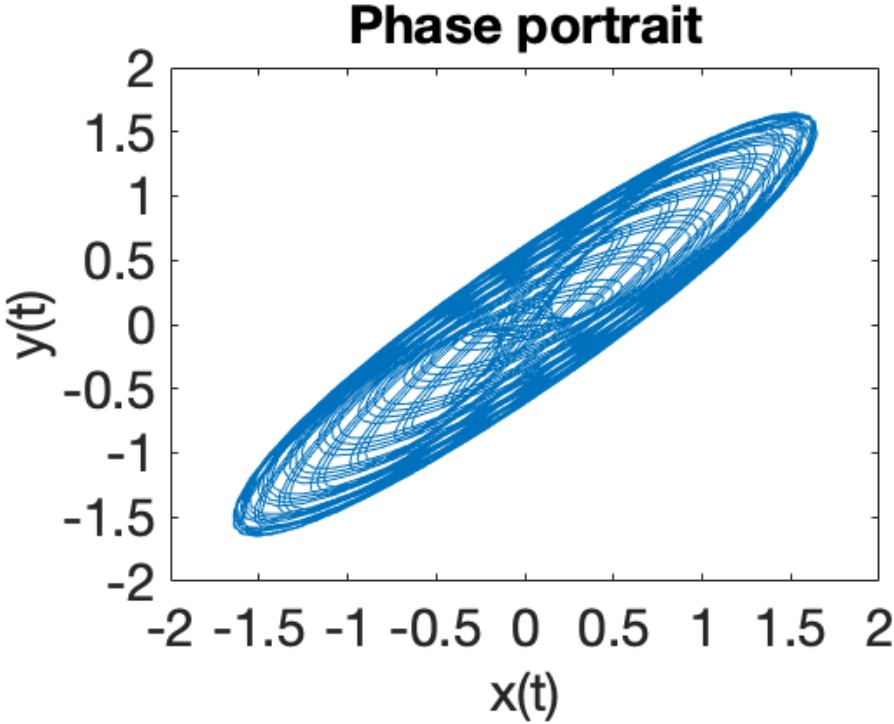

**Figure 2.** Parameter values $C_1 = 0.75, C_2 = 0.9, C_3 = 0.8, C_4 = 0.85, \varphi_1 = \pi/3, \varphi_2 = \pi/4, \varphi_3 = \pi/2, \varphi_4 = \pi/3, \omega_1/\omega_2 = 1.96/1.85 \longmapsto$ quasiperiodic dynamics.

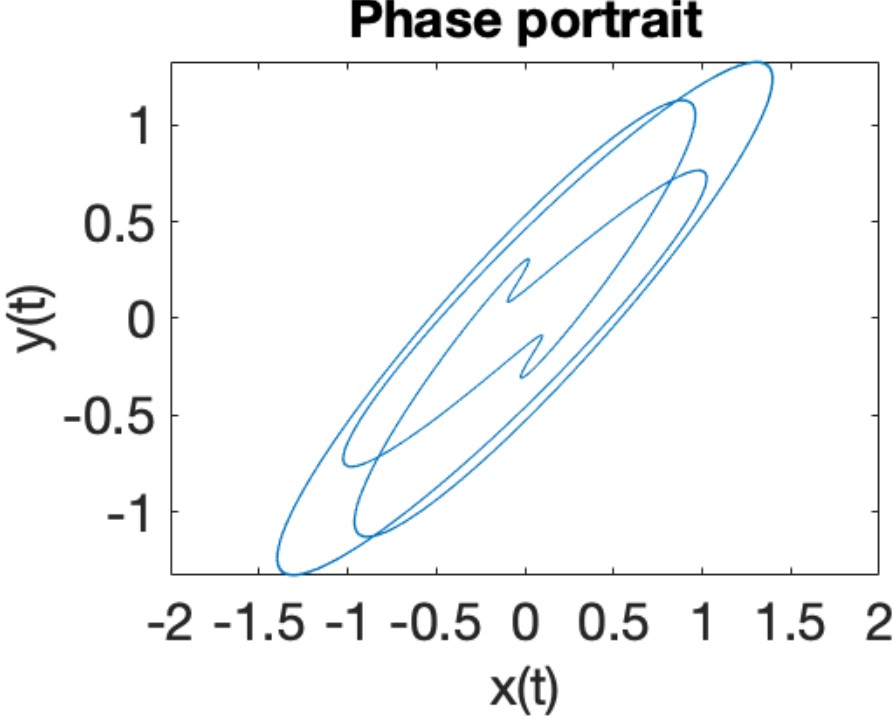

**Figure 3.** Parameter values identical with those from Figure 1, $\omega_1 = 1.2, \omega_2 = 2 \longmapsto$ periodic behavior.

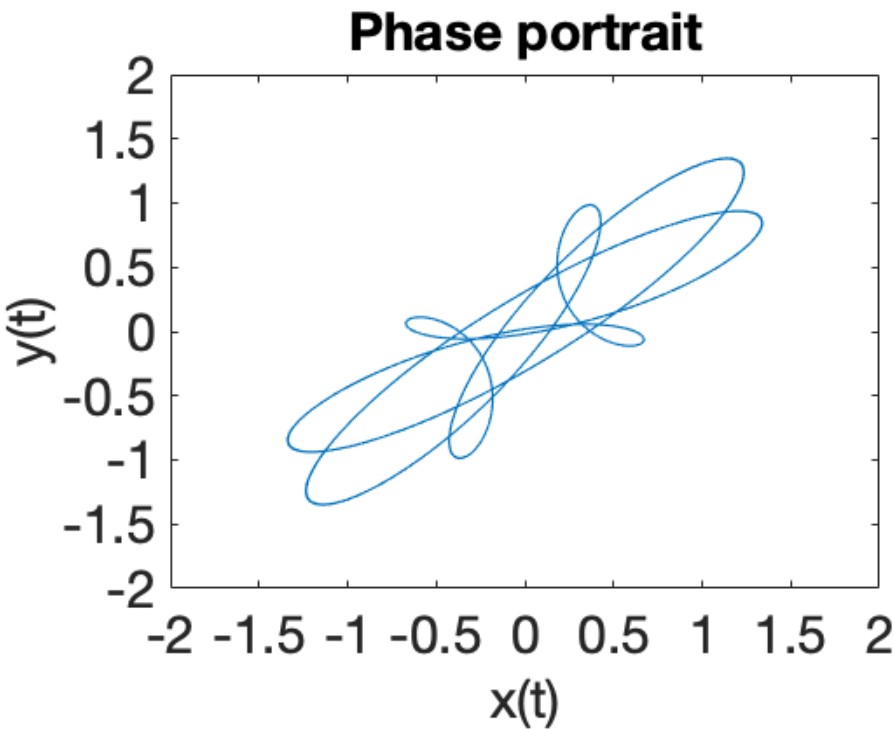

**Figure 4.** Parameter values $C_1 = 0.8, C_2 = 0.6, C_3 = 0.75, C_4 = 0.6, \varphi_1 = \pi/4, \varphi_2 = \pi/2, \varphi_3 = \pi/3,$ $\varphi_4 = \pi/5, \omega_1 = 1.2, \omega_2 = 2 \longmapsto$ periodic dynamics.

The phase portraits with parameter values $C_1 = 0.75, C_2 = 0.9, C_3 = 0.8, C_4 = 0.85$ are illustrated in Figures 5–8.

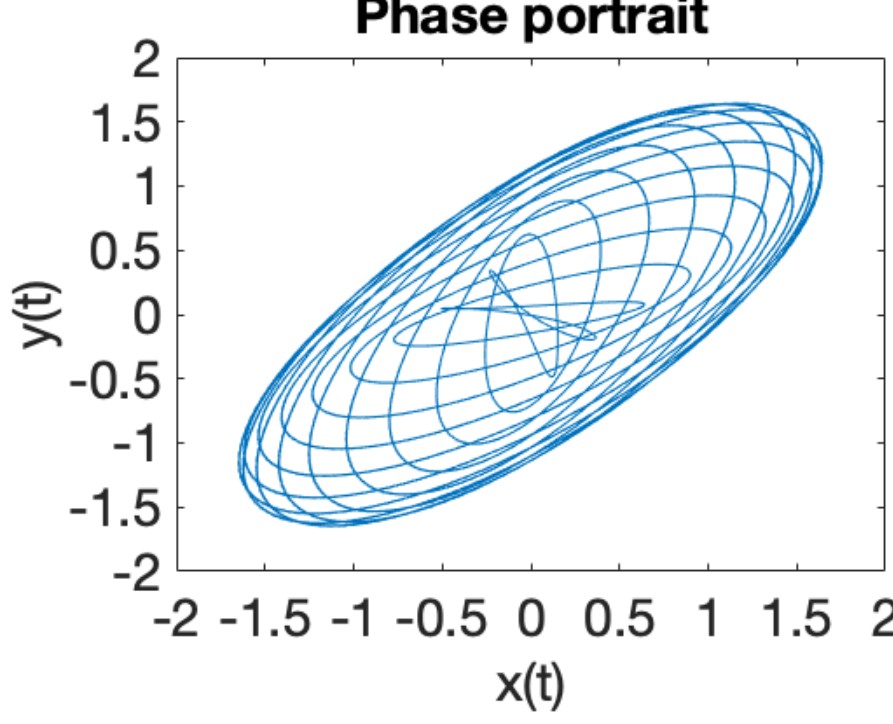

**Figure 5.** $\varphi_1 = \pi/5, \varphi_2 = \pi/6, \varphi_3 = \pi/3, \varphi_4 = \pi/2, \omega_1/\omega_2 = 1.8/1.7 \longmapsto$ periodic behavior.

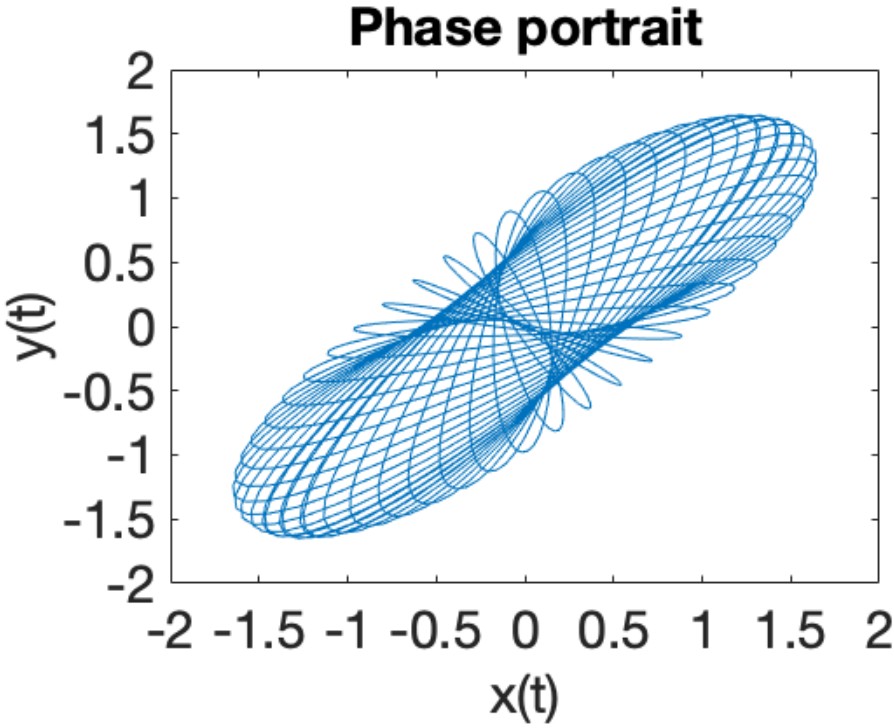

**Figure 6.** $\varphi_1 = \pi/4, \varphi_2 = \pi/2, \varphi_3 = \pi/3, \varphi_4 = \pi/4, \omega_1 = 1.81, \omega_2 = 1.88 \longmapsto$ quasiperiodic dynamics.

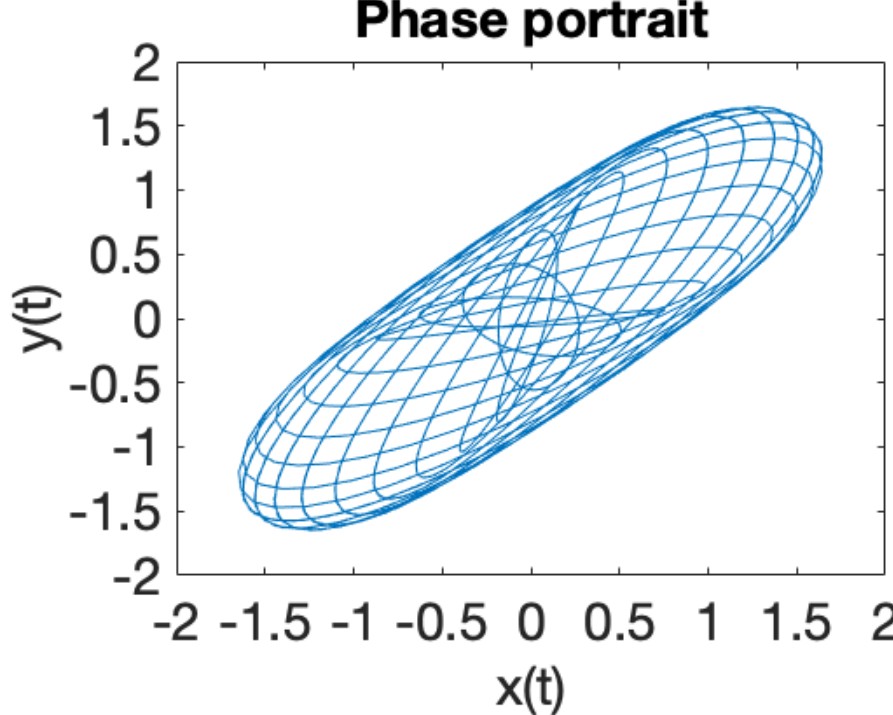

**Figure 7.** $\varphi_1 = \pi/6, \varphi_2 = \pi/5, \varphi_3 = \pi/2, \varphi_4 = \pi/4, \omega_1 = 1.8, \omega_2 = 1.9 \longmapsto$ periodic behavior.

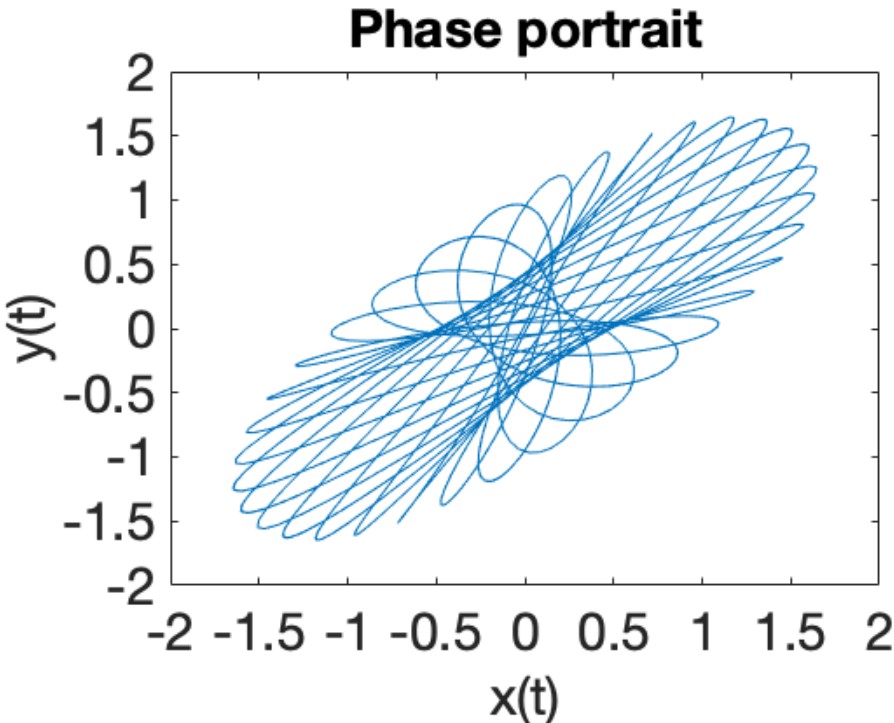

**Figure 8.** $\varphi_1 = \pi/6, \varphi_2 = \pi/4, \varphi_3 = \pi/2, \varphi_4 = \pi/8, \omega_1 = 1.5, \omega_2 = 1.9 \longmapsto$ periodic behavior.

Figure 1 illustrates the phase portrait for a system of two coupled oscillators in a RF trap, where the eigenfrequencies ratio is $\omega_1/\omega_2 = 1.71/1.93$. When the eigenfrequencies ratio is a rational number $\omega_1/\omega_2 \in \mathbb{Q}$, the solutions of the equations of motion (18) are periodic trajectories and ion dynamics is regular. In case when the eigenfrequencies ratio is an irrational number $\omega_1/\omega_2 \notin \mathbb{Q}$, iterative rotations occur around a certain point [40] that are called *ergodic*, according to the theorem of Weyl. It can be observed that the solutions (18) of the equations of motions, illustrated in Figures 1–8, demonstrate that ion dynamics is generally periodic or quasiperiodic, and stable. There are also a few cases which illustrate ergodic dynamics [40] and the occurrence of what may be interpreted as iterative rotations. According to Figures 1–8, by extending the motion in 3D we can ascertain that the trajectory executes periodic and quasiperiodic motion on the surface of a torus, referred to as a Kolmogorov–Arnold–Moser (KAM) torus [41]. By choosing various initial conditions different KAM tori can be generated.

We have integrated the equations of motion given by Equation (1) to explore ion dynamics and illustrate the associated power spectra [42], as shown in Figures 9–16. The numerical modeling we performed clearly demonstrates that ion dynamics is dominantly periodic or quasiperiodic.

Ref. [43] describes an electrodynamic ion trap in which the electric quadrupole field oscillates at two different frequencies. The authors report simultaneous tight confinement of ions with extremely different charge-to-mass ratios, e.g., singly ionized atomic ions together with multiply charged nanoparticles. The system represents the equivalent of two superimposed RF traps, where each one of them operates close to a frequency optimized in order to achieve tight storage for one of the species involved, which leads to strong and stable confinement for both particle species used.

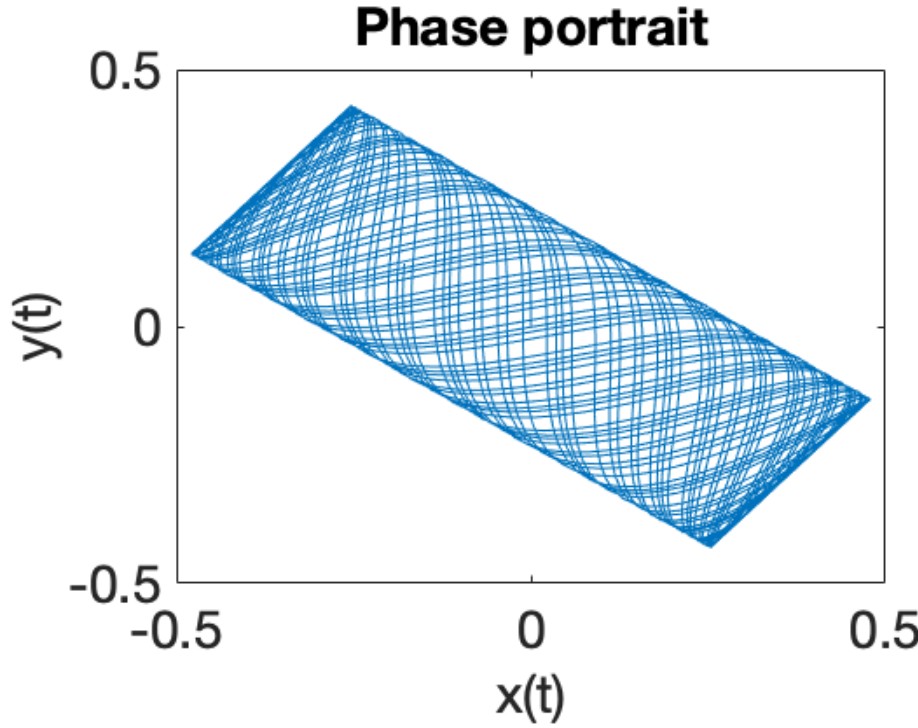

**Figure 9.** Phase portrait for $b = 2, k_1 = 4, k_2 = 5, m_1 = m_2 = 1$.

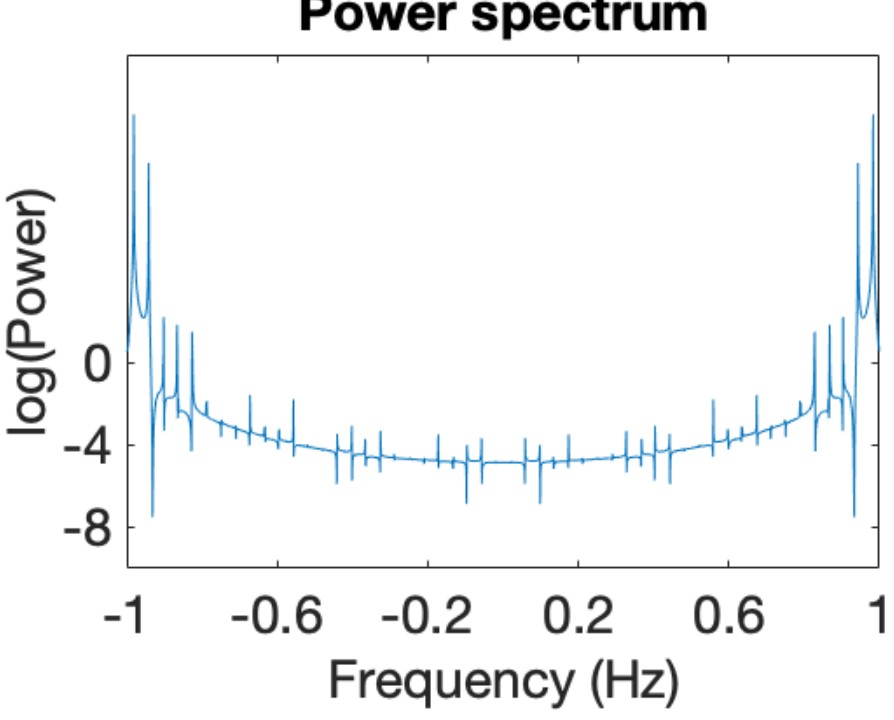

**Figure 10.** Associated power spectrum. Initial conditions $\dot{x}(0) = 0, \dot{y}(0) = 0, x(0) = 0, y(0) = 0.5$.

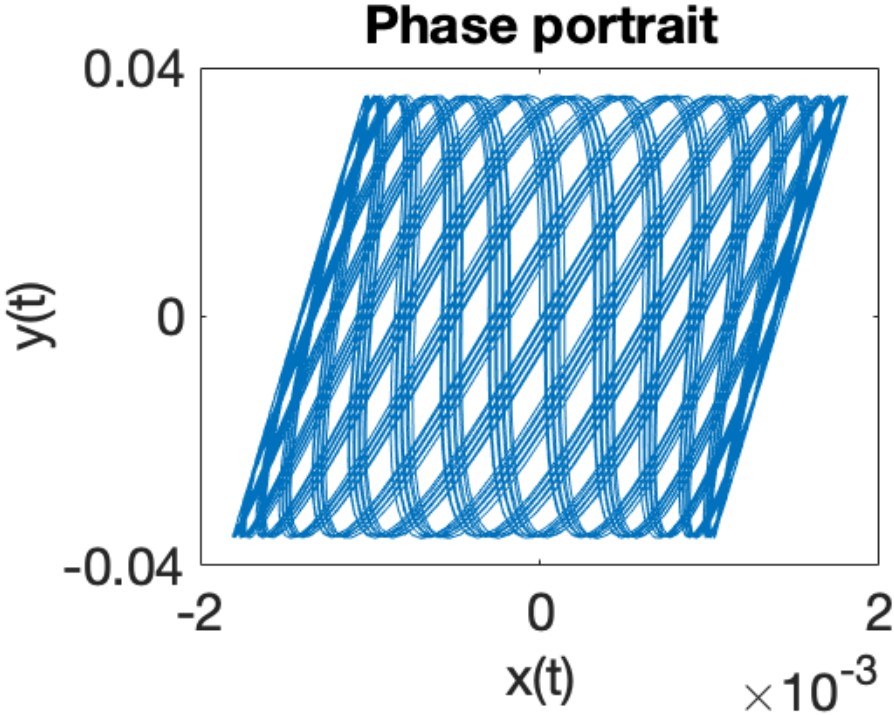

**Figure 11.** Phase portrait for $b = 2, k_1 = 17, k_2 = 199, m_1 = m_2 = 1$.

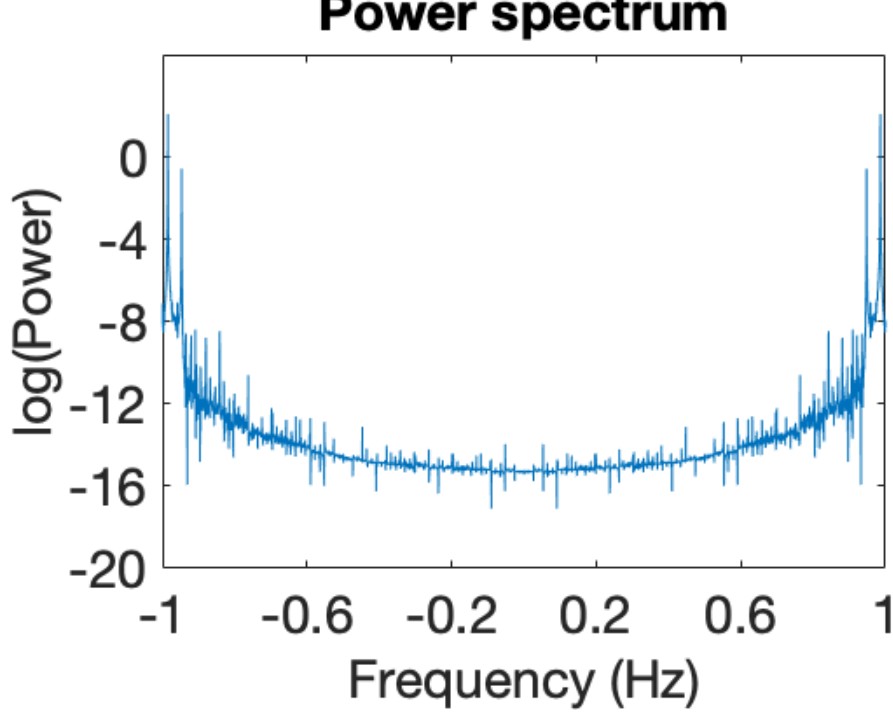

**Figure 12.** Associated power spectrum. Initial conditions $\dot{x}(0) = 0, \dot{y}(0) = 0, x(0) = 0, y(0) = 0.5$.

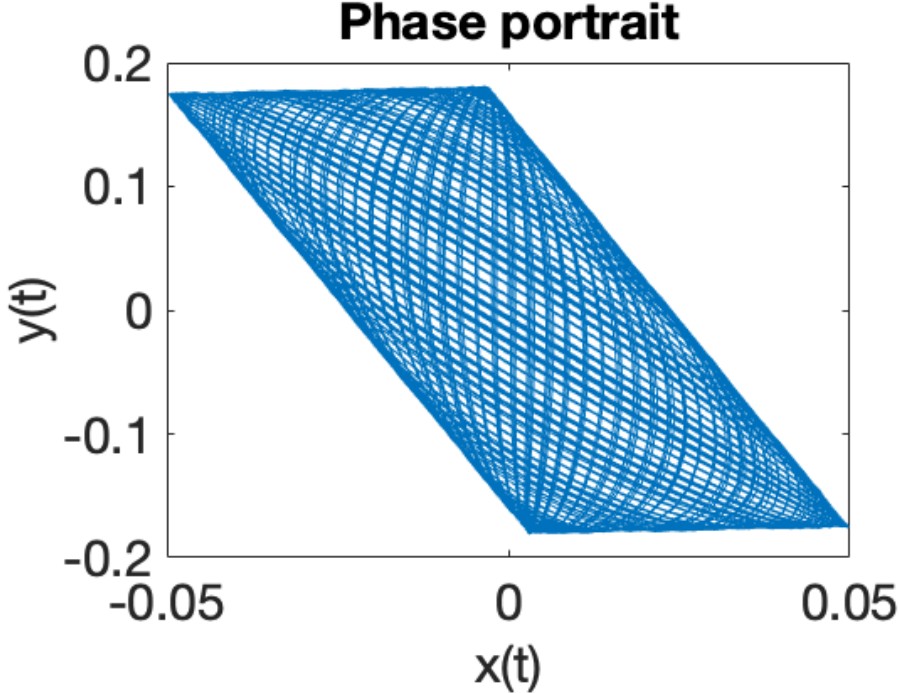

**Figure 13.** Phase portrait for $b = 3, k_1 = 99, k_2 = 102, m_1 = 10, m_2 = 13$.

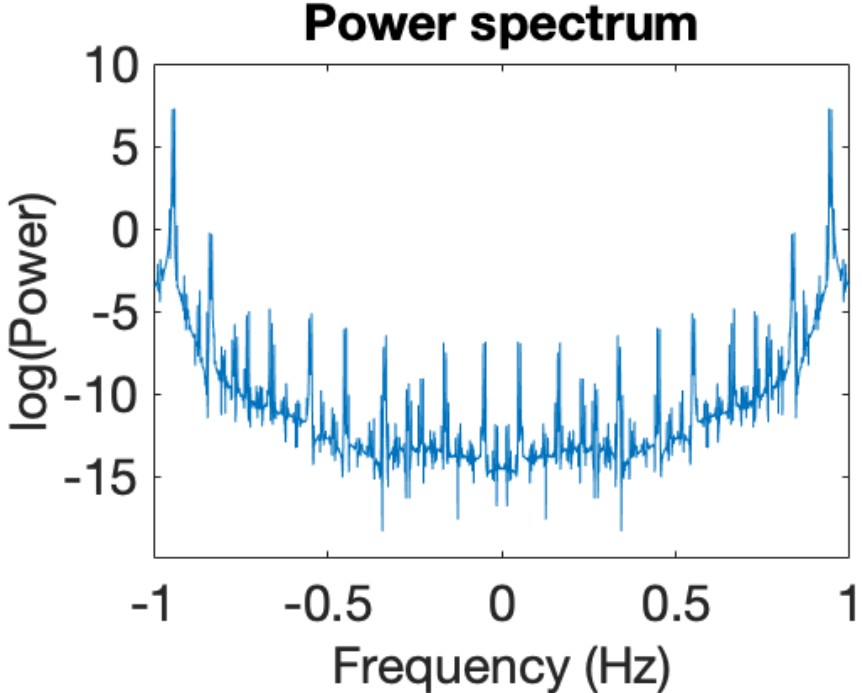

**Figure 14.** Associated power spectrum. Initial conditions $\dot{x}(0) = 0, \dot{y}(0) = 0, x(0) = 0, y(0) = 0.5$.

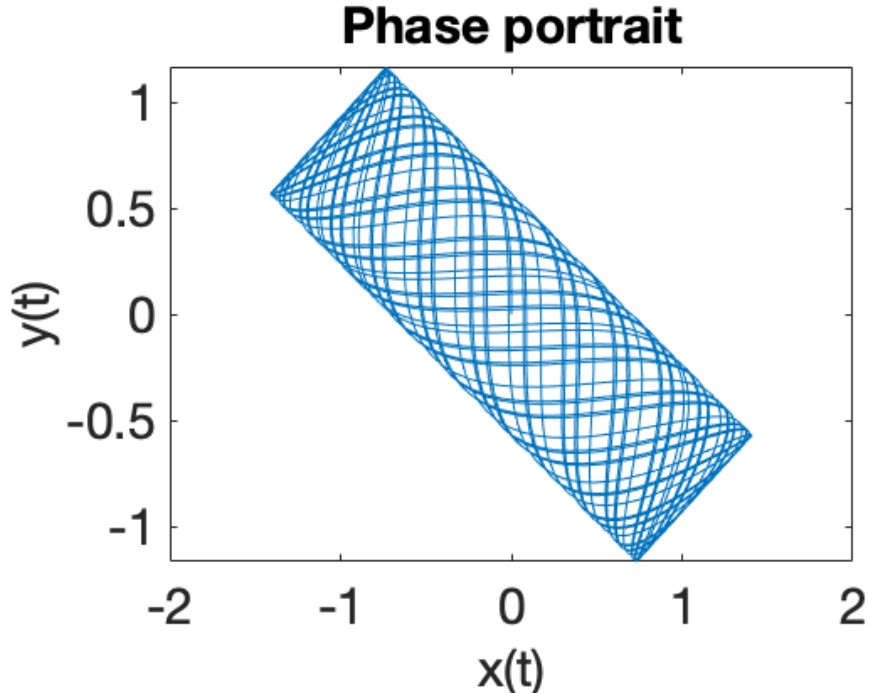

**Figure 15.** Phase portrait for $b = 2, k_1 = 4, k_2 = 5, m_1 = 5, m_2 = 7$.

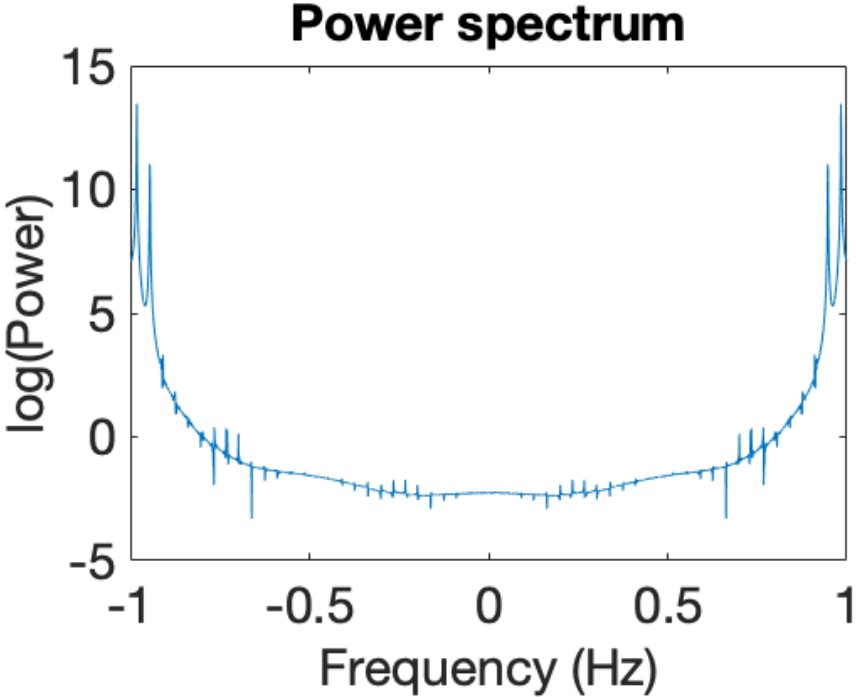

**Figure 16.** Associated power spectrum. Initial conditions $\dot{x}(0) = 0, \dot{y}(0) = 0, x(0) = 0, y(0) = 0.5$.

## 3. Dynamic Stability for Two Oscillators Levitated in a RF Trap

### 3.1. System Hamiltonian Hessian Matrix Approach

A well established model is employed to portray system dynamics, that relies on two control parameters: the axial angular moment and the ratio between the radial and axial

secular frequencies characteristic to the trap. If we consider two ions with equal electric charges, their relative motion is described by the equation [16,18,19]

$$
\frac{d^2}{dt^2}\begin{bmatrix} x \\ y \\ z \end{bmatrix} + [a + 2q\cos(2t)]\begin{bmatrix} x \\ y \\ -2z \end{bmatrix} = \frac{\mu_x^2}{|\mathbf{r}|^3}\begin{bmatrix} x \\ y \\ z \end{bmatrix}, \tag{23}
$$

where $\vec{r} = x_1 - x_2$, $\mu_x = \sqrt{a + \frac{1}{2}q^2}$ represents the dimensionless radial secular (pseudo-oscillator) characteristic frequency [44], while $a$ and $q$ stand for the adimensional trap parameters in the Mathieu equation, namely

$$
a = \frac{8QU_0}{m\Omega^2(r_0^2 + 2z_0^2)}, \quad q = \frac{4QV_0}{m\Omega^2(r_0^2 + 2z_0^2)}.
$$

$U_0$ and $V_0$ denote the d.c and RF trap voltage, respectively, $Q$ stands for the electric charge of the ion, $\Omega$ represents the RF drive frequency, while $r_0$ and $z_0$ are the trap radial and axial dimensions. For $a, q \ll 1$, such as in our case, the pseudopotential approximation is valid. Therefore we can associate an autonomous Hamilton function to the system described by Equation (23), which we express in scaled cylindrical coordinates $(\rho, \phi, z)$ as [16]

$$
H = \frac{1}{2}\left(p_\rho^2 + p_z^2\right) + U(\rho, z), \tag{24}
$$

where

$$
U(\rho, z) = \frac{1}{2}\left(\rho^2 + \lambda^2 z^2\right) + \frac{\nu^2}{2\rho^2} + \frac{1}{r}, \tag{25}
$$

with $r = \sqrt{\rho^2 + z^2}$, $\lambda = \mu_z/\mu_x$, and $\mu_z = \sqrt{2(q^2 - a)}$. $\nu$ denotes the scaled axial $(z)$ component of the angular momentum $L_z$ and it is a constant of motion, while $\mu_z$ represents the second secular frequency [16]. We emphasize that both $\lambda$ and $\nu$ are strictly positive control parameters. For arbitrary values of $\nu$ and for positive discrete values of $\lambda = 1/2, 1, 2$, Equation (25) is integrable and even separable, excluding the case when $\lambda = 1/2$ and $|\nu| > 0$ $(\nu \neq 0)$, as stated in [17].

The equations of the relative motion corresponding to the Hamiltonian function described by Equation (24) can be cast into [16,17]:

$$
\ddot{z} = \frac{z}{r^{3/2}} - \lambda^2 z,
$$

$$
\ddot{\rho} = \frac{\rho}{r^{3/2}} - \rho + \frac{\nu^2}{\rho^3}, \tag{26}
$$

with $p_\rho = \dot{\rho}$ and $p_z = \dot{z}$. The critical points of the $U$ potential are determined as solutions of the system of equations:

$$
\frac{\partial U}{\partial \rho} = \rho - \frac{\nu^2}{\rho^3} - \frac{1}{r^2}\frac{\rho}{r} = 0,
$$

$$
\frac{\partial U}{\partial z} = \lambda^2 z - \frac{1}{r^2}\frac{z}{r} = 0, \tag{27}
$$

where $\partial r/\partial \rho = \rho/r$ and $\partial r/\partial z = z/r$.

### 3.2. Solutions of the Equations of Motion for the Two Oscillator System

We use the Morse theory to determine the critical points of the potential $U$ and to discuss the solutions of Equation (27), with an aim to fully characterize the dynamical stability of the coupled oscillator system. Then

$$z\left(\lambda^2 - \frac{1}{r^3}\right) = 0, \tag{28}$$

which leads to a number of two possible cases:

**Case 1.** $z = 0$. The first equation of the system (27) can be rewritten as

$$\rho - \frac{v^2}{\rho^3} - \frac{\rho}{r^3} = 0,$$

which gives us $\rho = r$ for $z = 0$. In such case, a function results

$$f(\rho) = \rho^4 - \rho - v^2, \quad f'(\rho) = 4\rho^3 - 1. \tag{29}$$

The second relationship in Equation (29) shows that $\rho = \sqrt[3]{\frac{1}{4}}$ is a point of minimum for $f(\rho)$. In case when $\rho_0 > 0$, for $v \neq 0$ and $z_0 = 0$:

$$f(\rho) = \rho_0^4 - \rho_0 - v^2 = 0.$$

In case when $v = 0$ we obtain $f(\rho) = \rho(\rho^3 - 1) = 0$. Then, the solutions are $\rho_1 = 0$ and $\rho_2 = 1$, where only $\rho_2 = 1$ is a valid solution. Moreover, for $v = 0$ and $z_0 = 0$, $\rho_0 = 1$ is a solution.

**Case 2.** $r^3 = 1/\lambda^2 \Rightarrow$

$$r = \lambda^{-2/3}. \tag{30}$$

We return to the system of Equations (27) and infer

$$\rho = \frac{\sqrt{|v|}}{\sqrt[4]{1 - \lambda^2}}, \tag{31}$$

for $\lambda < 1$. In case when $\lambda \leq 1$ and $v \neq 0$, the system admits no solutions. In the scenario when $\lambda < 1, v = 0$ we find $\rho = 0$, while in case when $\lambda = 1$, $v = 0$ it results that any $\rho \geq 0$ represents a solution. As $r = \sqrt{z^2 + \rho^2}$, then

$$z = \pm\sqrt{r^2 - \rho^2} = \pm\sqrt{\lambda^{-4/3} - \rho^2}. \tag{32}$$

We differentiate among three possible sub-cases

*Subcase (i):* $\lambda < 1, v \neq 0$ and

$$z_{12} = \pm\sqrt{\lambda^{-4/3} - \frac{|v|}{\sqrt{1 - \lambda^2}}},$$

provided that $1 - \lambda^2 > v^2\lambda^{8/3}$ or

$$z = 0 \text{ for } 1 - \lambda_c^2 = v^2\lambda_c^{8/3},$$

where the $c$ index of $\lambda$ refers to critical.

*Subcase (ii):* $\lambda < 1, v = 0$ which leads to $\rho = 0$ and $z_{12} = \pm\lambda^{-2/3}$.

*Subcase (iii):* $\lambda = 1, v = 0$ which results in $z_{12} = \pm\sqrt{\lambda^{-4/3} - \rho^2}$, with $\rho \geq 0$.

These are the solutions we find for the equations of motion corresponding to the two coupled oscilators system. After doing the math, the Hessian matrix of the potential $U$ appears as

$$H = \begin{vmatrix} 1 + \frac{3v^2}{\rho^4} - \frac{1}{r^3} + \frac{3\rho^2}{r^5} & \frac{3\rho z}{r^5} \\ \frac{3\rho z}{r^5} & \lambda^2 - \frac{1}{r^3} + \frac{3z^2}{r^5} \end{vmatrix}. \tag{33}$$

The determinant and the trace of the Hessian matrix result as

$$\det H = \frac{3v^2}{\rho^4}\left(\lambda^2 - \frac{1}{r^3} + \frac{3z^2}{r^5}\right) + \lambda^2\left(1 - \frac{1}{r^3} + \frac{3\rho^2}{r^5}\right) - \frac{1}{r^3}\left(1 + \frac{2}{r^3} - \frac{3z^2}{r^2}\right),$$

$$\text{Tr}\, H = 1 + \lambda^2 + \frac{3v^2}{\rho^4} + \frac{1}{r^3}. \tag{34}$$

From Equation (34) we infer that $\text{Tr}\, H = 0$. Thus, the Hessian matrix $H$ has at least a strictly positive eigenvalue.

*3.3. Critical Points. Discussion.*

We use Equation (34) with an aim to investigate the critical points for the system of interest. We consider two distinct cases:
**Case 1.** $z = 0$ and $r = \rho$. Then Equations (34) modify appropriately

$$\det H = \frac{3v^2}{\rho^4}\left(\lambda^2 - \frac{1}{\rho^3}\right) + \lambda^2\left(1 + \frac{2}{\rho^3}\right) - \frac{1}{\rho^3}\left(1 + \frac{2}{\rho^3}\right),$$

$$\text{Tr}\, H = 1 + \lambda^2 + \frac{3v^2}{\rho^4} + \frac{1}{\rho^3}. \tag{35}$$

We discriminate among the following sub-cases:
*Subcase (i): $v = 0$, $z = 0$, $\rho = 1$.*
We obtain a system of equations as follows

$$\text{Tr}\, H = 2 + \lambda^2 > 0,$$

$$\det H = 3\left(\lambda^2 - 1\right). \tag{36}$$

Moreover, we infer a table (see Table 1) that describes the eigenvalues $\lambda_1$ and $\lambda_2$ of the Hessian matrix:

**Table 1.** Hessian matrix eigenvalues and associated critical points.

|  | $\lambda_1$ | $\lambda_2$ | **Critical Point** |
|---|---|---|---|
| $0 < \lambda < 1$ | $> 0$ | $> 0$ | Minimum |
| $\lambda = 1$ | $> 0$ | $0$ | Degeneracy |
| $\lambda > 1$ | $> 0$ | $< 0$ | Saddle point |

Thus, by investigating the signs of the Hessian matrix eigenvalues we can discriminate between critical points, minimum points, saddle points and degeneracy. Only if $\lambda = 1$ the critical point is degenerate. When the determinant of the Hessian matrix of the potential $\det H \neq 0$ the system is non-degenerate. The system is degenerate if $\det H = 0$.
*Subcase (ii): $z = 0$, $v \neq 0$.*

The derivatives of a smooth function must be continuous. We now turn back to Equation (29). Then $v^2 = \rho(\rho^3 - 1)$ and

$$\det \mathsf{H} = \left(4 - \frac{1}{\rho^3}\right)\left(\lambda^2 - \frac{1}{\rho^3}\right) . \tag{37}$$

We seek for degenerate critical points (characterized by $\det \mathsf{H} = 0$). We infer $\rho = \lambda^{-2/3}$ or $\rho = 4^{-3}$, which involves two distinct sub-subcases:

(a) $\rho = \lambda^{-2/3}$. We return to Equation (27) and infer

$$\lambda^{-8/3} - \lambda^{-2/3} - v^2 = 0 .$$

(b) $\rho^4 \geq \rho$ , $\rho \geq 1$. In such a situation we encounter a point of minimum when $\rho > \lambda^{-2/3}$, while the case $\rho < \lambda^{-2/3}$ implies a saddle point.

**Case 2.** $r = \lambda^{-4/3}$ , $z^2 = \lambda^{-4/3} - \rho^2$ .

In this particular situation, after doing the math Equation (34) can be recast into

$$\det \mathsf{H} = 12\lambda^2\left(1 - \lambda^2\right)\left(1 - \lambda^{4/3}\rho^2\right) , \tag{38}$$
$$\mathrm{Tr}\,\mathsf{H} = 4 - \lambda^2 > 0 ,$$

with $0 \leq \lambda^2 \leq 1$. We differentiate among several sub-cases as follows:

*Subcase (i):* $v = 0$ , $\lambda^2 = 1$. We further infer $z = \pm\sqrt{\lambda^{-4/3} - \rho^2}$, with $\rho^2 \in \left[-\lambda^{2/3}, \lambda^{2/3}\right]$, $\rho \leq -\lambda^{-8/3}$. Then $\mathrm{Tr}\,\mathsf{H} = 3$ and $\det \mathsf{H} = 0$, which characterizes a degenerate critical point.

*Subcase (ii):* $v \neq 0$ , $\lambda^2 = 0$. We are in the case of a degenerate critical point, with $\rho = \sqrt{|v|}$. In this particular case $\det \mathsf{H} = 0$ and

$$v^2 = \lambda^{-8/3} - \lambda^{-2/3} .$$

*Subcase (iii):* $0 \leq \lambda^2 < 1$ , $v \neq 0$. The critical point is a point of minimum, as $\det \mathsf{H} > 0$:

$$\rho_0 = \frac{\sqrt{|v|}}{\sqrt[4]{1 - \lambda^2}} , \quad 1 - \lambda^2 > v^2\lambda^{4/3} ,$$

$$z_{1,2} = \sqrt{\lambda^{-4/3} - \frac{v}{\sqrt{1 - \lambda^2}}} .$$

*Subcase (iv):* $0 < \lambda^2 < 1$ , $v = 0$. Then $\rho = 0$ and $\det \mathsf{H} = 0$, which indicates a point of minimum characterized by $z_{12} = \pm\lambda^{-2/3}$.

*Subcase (v):* Case $v = 0$ , $\lambda = 0$. In such case we infer $\rho = 0$ , $z = 0$. We are in the case of a degenerate critical point as $(\det \mathsf{H} = 0)$.

*Subcase (vi):* Case $\rho = \lambda^{-2/3}$, $\lambda = \lambda_c$.

$$1 - \lambda_c^2 = v^2\lambda_c^{8/3} . \tag{39}$$

The critical point is degenerate with $z = 0$ and $(\det \mathsf{H} = 0)$.

A critical point for which the Hessian matrix is non-singular, is called a non-degenerate critical point. A Morse function admits only non-degenerate critical points that are stable [45]. The degenerate critical points (defined by $\det \mathsf{H} = 0$) compose the bifurcation set, whose image in the control parameter space (more precisely the $v - \lambda$ plan) establishes the catastrophe set of equations that defines the separatrix:

$$v = \sqrt{\lambda^{-8/3} - \lambda^{-2/3}} \quad \text{or} \quad \lambda = 0 . \tag{40}$$

Our method relies on employing the Hessian matrix to better characterize dynamical stability and the critical points of the system. Figure 17 displays the bifurcation

diagram for two coupled oscillators confined in a Paul trap. The ion relative motion is characterized by the Hamilton function described by Equations (24) and (25). The diagram illustrates both stability and instability regions where ion dynamics is integrable and non-integrable, respectively. Ion dynamics is integrable and even separable when $\lambda = 0.5$, $\lambda = 1$, $\lambda = 2$ [17–19].

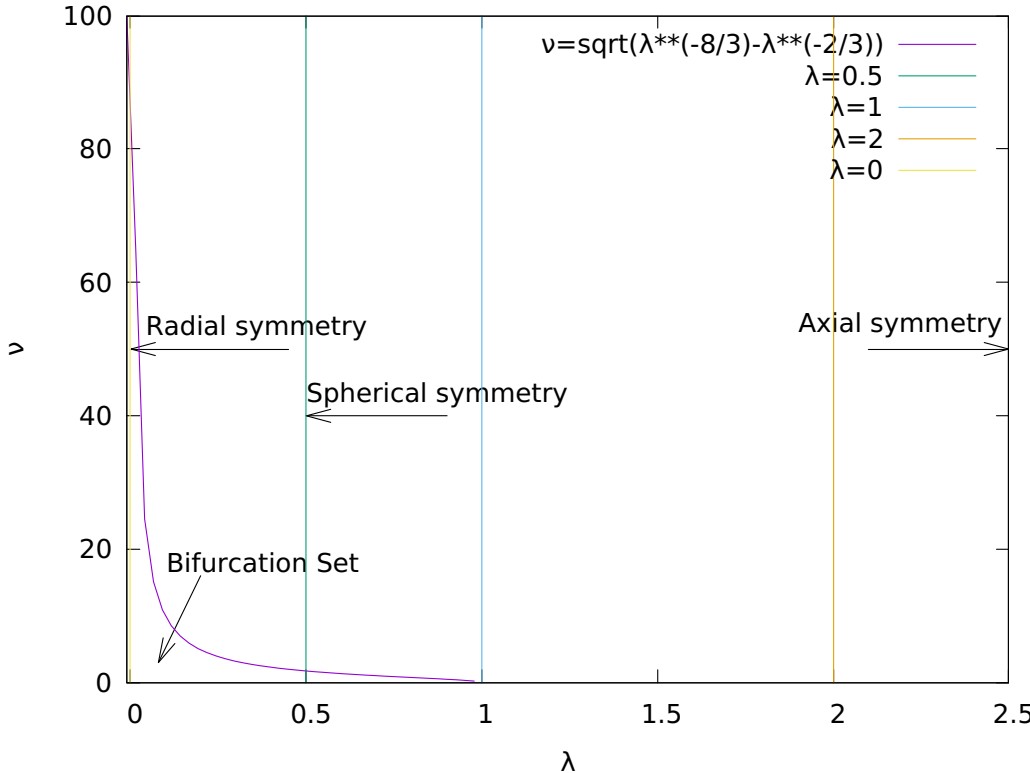

**Figure 17.** The bifurcation set for a system of two ions confined in a Paul trap.

## 4. Quantum Stability and Ordered Structures for Many-Body Systems of Trapped Ions

Furthermore, we apply the Hessian matrix approach and method previously introduced to investigate semiclassical stability and ordered structures for strongly coupled Coulomb systems (SCCS) confined in a 3D QIT. In addition we suggest an analytical a method to determine the associated critical points. We consider a system consisting of $N$ identical ions of mass $m_\alpha$ and electric charge $Q_\alpha$ ($\alpha = 1, 2, \ldots, N$), confined in a 3D RF (Paul) trap. The coordinate vector of the particle labeled as $\alpha$ is denoted by $\vec{r}_\alpha = (x_\alpha, y_\alpha, z_\alpha)$. A number of $3N$ generalized quantum coordinates $q_{\alpha i}, i = 1, 2, 3$, are associated to the $3N$ degrees of liberty. We also denote $q_{\alpha 1} = x_\alpha$, $q_{\alpha 2} = y_\alpha$, and $q_{\alpha 3} = z_\alpha$. Hence, the kinetic energy for a number of $\alpha$ particles confined in the trap can be expressed as

$$T = \sum_{\alpha=1}^{N} \sum_{i=1}^{3} \frac{1}{2m_\alpha} \dot{q}_{\alpha i}^2 \, , \tag{41}$$

while the potential energy is

$$U = \frac{1}{2} \sum_{\alpha=1}^{N} \sum_{i=1}^{3} k_i q_{\alpha i}^2 + \sum_{1 \leq \alpha < \beta \leq N} \frac{Q_\alpha Q_\beta}{4\pi \varepsilon_0 |\vec{r}_\alpha - \vec{r}_\beta|} \, , \tag{42}$$

where $\varepsilon_0$ stands for the vacuum permittivity. For a spherical 3D trap: $k_1 = k_2 = k_3$. In case of Paul or Penning 3D QIT $k_1 = k_2$, while the linear 2D Paul trap (LIT) corresponds to

$k_3 = 0$, $k_1 = -k_2$. The $k_i$ constants in case of a Paul trap result from the pseudopotential approximation. The critical points of the system result as:

$$\sum_{\alpha=1}^{N} \Delta_\alpha U = 0 \, , \; \Delta_\alpha = \frac{\partial^2}{\partial x_\alpha^2} + \frac{\partial^2}{\partial y_\alpha^2} + \frac{\partial^2}{\partial z_\alpha^2} \, , \tag{43}$$

$$\Delta U = 0 \text{ or } \frac{\partial U}{\partial q_{\gamma j}} = 0, \; \gamma = 1, \dots, N; \; j = 1, 2, 3. \tag{44}$$

We denote

$$\frac{\partial q_{\alpha i}}{\partial q_{\gamma j}} = \delta_{\alpha\gamma} \delta_{ij} \, , \tag{45}$$

where $\delta$ stands for the Kronecker delta function. After doing the math we can write Equation (44) as

$$\frac{\partial U}{\partial q_{\gamma j}} = \frac{1}{2} \sum_{\alpha=1}^{N} \sum_{i=1}^{3} 2 k_i q_{\alpha i} \delta_{\alpha\gamma} \delta_{ij} - \sum_{1 \le \alpha < \beta \le N} \frac{1}{4\pi\varepsilon_0} \frac{Q_\alpha Q_\beta}{|\vec{r}_\alpha - \vec{r}_\beta|^3} \left( q_{\alpha j} - q_{\beta j} \right) \left( \delta_{\alpha\gamma} - \delta_{\beta\gamma} \right) , \tag{46}$$

where the second term in Equation (46) represents the energy of the system, and introduce

$$\xi_{\alpha\beta} = \frac{1}{4\pi\varepsilon_0} \frac{Q_\alpha Q_\beta}{|\vec{r}_\alpha - \vec{r}_\beta|^3}, \; \alpha \neq \beta. \tag{47}$$

Moreover, $\xi_{\alpha\beta} = \xi_{\beta\alpha}$. After some calculus the system energy can be cast as

$$E = q_{\gamma j} \sum_{\alpha=1}^{N} \xi_{\alpha\gamma} - \sum_{\alpha=1}^{N} \xi_{\alpha\gamma} q_{\alpha j} \, . \tag{48}$$

We use Equations (46) and (48), while the critical points (in particular, the minima) result as a solution of the system of equations

$$\frac{\partial U}{\partial q_{\gamma j}} = \left( k_j - \sum_{\alpha=1}^{N} \xi_{\alpha\gamma} \right) q_{\gamma j} + \sum_{\alpha=1}^{N} \xi_{\alpha\gamma} q_{\alpha j} = 0, \; 1 \le j \le 3, \; 1 \le \alpha \le N \, . \tag{49}$$

We consider $\bar{q}_{\alpha j}$ to be a solution of the system of Equation (49) and obtain

$$U(q) = U(\bar{q}) + \sum_{\alpha=1}^{N} \sum_{j=1}^{3} \frac{\partial U}{\partial q_{\alpha j}} \left( q_{\alpha j} - \bar{q}_{\alpha j} \right) +$$
$$\frac{1}{2} \sum_{\alpha,\alpha'=1}^{N} \sum_{j,j'=1}^{3} \frac{\partial^2 U}{\partial q_{\alpha j} \partial q_{\alpha' j'}} \left( q_{\alpha j} - \bar{q}_{\alpha j} \right) \left( q_{\alpha' j'} - \bar{q}_{\alpha' j'} \right) + \dots \, . \tag{50}$$

We further infer

$$\frac{\partial^2 U}{\partial q_{\gamma j} \partial q_{\gamma' j'}} = k_j \delta_{\gamma\gamma'} \delta_{jj'} - \sum_{\alpha=1}^{N} \xi_{\alpha\gamma} \delta_{\gamma\gamma'} \delta_{jj'} - q_{\gamma j} \sum_{\alpha=1}^{N} \frac{\partial \xi_{\alpha\gamma}}{\partial q_{\gamma' j'}} + \sum_{\alpha=1}^{N} \xi_{\gamma\gamma'} \delta_{jj'} + \sum_{\alpha=1}^{N} \frac{\partial \xi_{\alpha\gamma}}{\partial q_{\gamma' j'}} q_{\alpha j} \, . \tag{51}$$

After performing the math (details are supplied in Appendix C) we cast Equation (51) into

$$\frac{\partial^2 U}{\partial q_{\gamma j} \partial q_{\gamma' j'}} = - \left( \sum_{\alpha=1}^{N} \xi_{\alpha\gamma} \right) \delta_{\gamma\gamma'} \delta_{jj'} + \xi_{\gamma\gamma'} \delta_{jj'} + k_j \delta_{\gamma\gamma'} \delta_{jj'} +$$
$$q_{\gamma j} \sum_{\alpha=1}^{N} \eta_{\alpha\gamma} \left( q_{\alpha j'} - q_{\gamma j'} \right) \left( \delta_{\alpha\gamma'} - \delta_{\gamma\gamma'} \right) - \sum_{\alpha=1}^{N} \eta_{\alpha\gamma} \left( q_{\alpha j'} - q_{\gamma j'} \right) \left( \delta_{\alpha\gamma'} - \delta_{\gamma\gamma'} \right) q_{\alpha j} \, . \tag{52}$$

We search for a fixed solution $q^0_{\gamma j}$ of the system of Equation (49). Then, the elements of the Hessian matrix of the potential function $U$ in a critical point of coordinates $q^0_{\gamma j}$ are:

$$
\frac{\partial^2 U}{\partial q^0_{\gamma j} \partial q^0_{\gamma' j'}} = k_j \delta_{\gamma\gamma'} \delta_{jj'} + \xi_{\gamma\gamma'} \delta_{jj'} - \left( \sum_{\alpha=1}^{N} \xi_{\alpha\gamma} \right) \delta_{\gamma\gamma'} \delta_{jj'} +
$$
$$
\eta_{\gamma\gamma'} \left( q^0_{\gamma j} q^0_{\gamma' j'} - q^0_{\gamma j} q^0_{\gamma j'} - q^0_{\gamma' j} q^0_{\gamma' j'} + q^0_{\gamma' j} q^0_{\gamma j'} \right) + q^0_{\gamma j} q^0_{\gamma j'} \sum_{\alpha=1}^{N} \eta_{\alpha\gamma} \delta_{\gamma\gamma'} +
$$
$$
\delta_{\gamma\gamma'} \sum_{\alpha=1}^{N} \eta_{\alpha\gamma} q^0_{\alpha j} q^0_{\alpha j'} - \delta_{\gamma\gamma'} q^0_{\gamma j} \sum_{\alpha=1}^{N} \eta_{\alpha\gamma} q^0_{\alpha j'} - \delta_{\gamma\gamma'} q^0_{\gamma j'} \sum_{\alpha=1}^{N} \eta_{\alpha\gamma} q^0_{\alpha j} . \quad (53)
$$

As it can be observed from Equation (53), our method allows one to determine (identify) the critical points of the potential function for the quantum system of $N$ identical ions, where equilibrium configurations occur. It is exactly these equilibrium configurations that present a large interest for ion crystals or for quantum logic.

## 5. Hamiltonians for Systems of $N$ Ions

We further apply our model to explore dynamical stability for systems consisting of $N$ identical ions confined in a 3D QIT (Paul, Penning or combined traps) and show they can be studied locally in the neighbourhood of the minimum configurations that describe ordered structures (Coulomb or ion crystals [46]). Collective dynamics for many body systems confined in a 3D QIT that exhibits cylindrical (axial) symmetry is characterized in Refs. [1,22]. We explore a system consisting of $N$ ions in a space with $d$ dimensions, labeled as $\mathbb{R}^d$. The coordinates in the manifold of configurations $\mathbb{R}^d$ are denoted by $x_{\alpha j}$, $\alpha = 1, \ldots, N$, $j = 1, \ldots, d$. In case of linear, planar or 3D (space) models, the number of corresponding dimensions is $d = 1$, $d = 2$ or $d = 3$, respectively. We will further introduce the kinetic energy $T$, the linear potential energy $U_1$, the 3D QIT potential energy $U$, and the anharmonic trap potential $V$:

$$
T = \sum_{\alpha=1}^{N} \sum_{j=1}^{d} \frac{1}{2m_\alpha} p^2_{\alpha j} , \quad U_1 = \frac{1}{2} \sum_{\alpha=1}^{N} \sum_{j=1}^{d} \delta_j x_{\alpha j} ,
$$
$$
U = \frac{1}{2} \sum_{\alpha=1}^{N} \sum_{i,j=1}^{d} \kappa_{ij} x^2_{\alpha j} , \quad V = \sum_{\alpha=1}^{N} v(\mathbf{x}_\alpha, t) ,
$$

(54)

where $m_\alpha$ is the mass of an ion labeled by $\alpha$, $\mathbf{x}_\alpha = (x_{\alpha 1}, \ldots, x_{\alpha d})$, while $\delta_j$ and $\kappa_{ij}$ represent functions that ultimately depend on time. The Hamilton function associated to the strongly coupled Coulomb system (SCCS) under investigation is

$$
H = T + U_1 + U + V + W ,
$$

where $W$ denotes the interaction potential between the ions.

Under the assumption of equal ion masses we introduce $d$ coordinates $x_j$ of the center of mass (CM)

$$
x_j = \frac{1}{N} \sum_{\alpha=1}^{N} x_{\alpha j} ,
$$

(55)

and $d(N-1)$ coordinates $y_{\alpha j}$ to account for the relative ion motion

$$
y_{\alpha j} = x_{\alpha j} - x_j , \quad \sum_{\alpha=1}^{N} y_{\alpha j} = 0 .
$$

(56)

We also introduce $d$ collective coordinates $s_j$ and the collective coordinate $s$, identified as

$$s_j = \sum_{\alpha=1}^{N} y_{\alpha j}^2 \ , \quad s = \sum_{\alpha=1}^{N} \sum_{j=1}^{d} y_{\alpha j}^2 \ . \tag{57}$$

Then

$$\sum_{\alpha=1}^{N} x_{\alpha j}^2 = N x_j^2 + \sum_{\alpha=1}^{N} y_{\alpha j}^2 \ . \tag{58}$$

$$s_j = \frac{1}{2N} \sum_{\alpha,\beta=1}^{N} \left( x_{\alpha j} - x_{\beta j} \right)^2 \ , \quad s = \frac{1}{2N} \sum_{\alpha,\beta=1}^{N} \sum_{j=1}^{d} \left( x_{\alpha j} - x_{\beta j} \right)^2 \ . \tag{59}$$

Equation (59) shows $s$ to symbolize the squared distance measured between the origin (fixed in the CM) and the point that designates the system of $N$ ions in the manifold of configurations. The relation $s = s_0$, with $s_0 > 0$ constant, establishes a sphere of radius $\sqrt{s_0}$ whose centre is located in the origin (of the configurations manifold). When investigating ordered structures of $N$ ions, the trajectory is restricted within a neighbourhood $\|s - s_0\| < \varepsilon$ of this sphere, with $\varepsilon$ sufficiently small. At the same time, the collective variable $s$ can be also regarded as a dispersion:

$$s = \sum_{\alpha=1}^{N} \sum_{j=1}^{d} \left( x_{\alpha j}^2 - x_j^2 \right) \ . \tag{60}$$

We now submit $p_{\alpha j}$ moments associated to the coordinates $x_{\alpha j}$. We also introduce $d$ moments $p_j$ of the center of mass (CM) and $d(N-1)$ moments $\xi_{\alpha j}$ of the relative ion motion defined as

$$p_j = \frac{1}{N} \sum_{\alpha=1}^{N} p_{\alpha j} \ , \quad \xi_{\alpha j} = p_{\alpha j} - p_j \ , \quad \sum_{\alpha=1}^{N} \xi_{\alpha j} = 0 \ , \tag{61}$$

with $p_{\alpha j} = -i\hbar \left( \partial / \partial x_{\alpha j} \right)$. We denote

$$D_j = \frac{1}{N} \sum_{\alpha=1}^{N} \frac{\partial}{\partial x_{\alpha j}} \ , \quad D_{\alpha j} = \frac{\partial}{\partial x_{\alpha j}} - D_j \ , \quad \sum_{\alpha=1}^{N} D_{\alpha j} = 0 \ . \tag{62}$$

In addition

$$\sum_{\alpha=1}^{N} \frac{\partial^2}{\partial x_j^2} = N D_j^2 + \sum_{\alpha=1}^{N} D_{\alpha j}^2 \ . \tag{63}$$

When $d = 3$ we denote by $L_{\alpha 3}$ the projection of the angular momentum of the $\alpha$ particle on axis 3. Then, the projections of the total angular momentum and of the relative motion angular momentum on axis 3, are labeled as $L_3$ and $L_3'$ respectively, determined as

$$\sum_{\alpha=1}^{N} L_{\alpha 3} = L_3 + L_3' \ , \quad L_{\alpha 3} = x_{\alpha 1} p_{\alpha 2} - x_{\alpha 2} p_{\alpha 1} \ ,$$

$$L_3 = p_1 D_2 - p_2 D_1 \ , \quad L_3' = \sum_{\alpha=1}^{N} \left( y_{\alpha 1} \xi_{\alpha 2} - y_{\alpha 2} \xi_{\alpha 1} \right) \ . \tag{64}$$

The Hamilton function assigned to a many body system of $N$ charged particles of mass $M$ and equal electric charge $Q$, confined in a quadrupole combined (Paul and Penning)

trap that displays axial (cylindrical) symmetry, in presence of a constant axial magnetic field $B_0$, can be expressed as [1,36]

$$H = \sum_{\alpha=1}^{N} \left[ \frac{1}{2M} \sum_{j=1}^{3} p_{\alpha j}^2 + \frac{K_r}{2} \left( x_{\alpha 1}^2 + x_{\alpha 2}^2 \right) + \frac{K_a}{2} x_{\alpha 3}^2 - \frac{\omega_c}{2} L_{\alpha 3} \right] + W \, ,$$

with

$$K_r = \frac{M\omega_c^2}{4} - 2Qc_2 A(t) \, , \; K_a = 4Qc_2 A(t) \, , \; \omega_c = \frac{QB_0}{M} \, ,$$

where $\omega_c$ is the cyclotron frequency characteristic to a Penning trap, $c_2$ is a constant that depends on the trap geometry and $A(t)$ represents a time periodical function [47]. The index $r$ refers to radial motion while the index $a$ refers to axial motion. We can also write $H$ by adding the Hamilton function of the CM, $H_{CM}$, and the Hamilton function associated to the ion relative motion $H'$:

$$H = H_{CM} + H' \, ,$$

$$H_{CM} = \frac{1}{2NM} \sum_{j=1}^{3} p_j^2 + \frac{NK_r}{2} \left( x_1^2 + x_2^2 \right) + \frac{NK_a}{2} x_3^2 - \frac{\omega_c}{2} L_3 \, ,$$

$$H' = \sum_{\alpha=1}^{N} \left[ -\frac{\hbar^2}{2M} \sum_{j=1}^{3} \xi_{\alpha j}^2 + \frac{K_r}{2} \left( y_{\alpha 1}^2 + y_{\alpha 2}^2 \right) + \frac{K_a}{2} y_{\alpha 3}^2 \right] - \frac{\omega_c}{2} L_3' + W \, .$$

(65)

Our results are in agreement with Ref. [22], where collective dynamical systems associated to the symplectic group are used to describe the axial and radial quantum Hamiltonians of the CM and of the relative ion motion. The space charge and its effect on the ion dynamics in case of a LIT is examined in Ref. [48], where the authors emphasize two distinguishable effects: (i) alteration of the specific ion oscillation frequency owing to variations of the trap potential, and (ii) for specific high charge density experimental conditions, the ions might perform as a single collective ensemble and exhibit dynamic frequency which is autonomous with respect to the number of ions. The model we suggest in this paper is appropriate to achieve a unitary approach aimed at generalizing the parameters for different types of 3D QIT. Further on, we apply this model to investigate the particular case of a combined Paul and Penning 3D QIT [49].

We consider $W$ to be an interaction potential that is translation invariant (it only depends of $y_{\alpha j}$). The ion distribution in the trap can be represented by means of numerical analysis and computer modeling [50,51], through the Hamilton function we provide

$$H_{sim} = \sum_{i=1}^{n} \frac{1}{2M} p_i^2 + \sum_{i=1}^{n} \frac{M}{2} \left( \omega_1^2 x_i^2 + \omega_2^2 y_i^2 + \omega_3^2 z_i^2 \right)$$
$$+ \sum_{1 \le i < j \le n} \frac{Q^2}{4\pi\varepsilon_0} \frac{1}{|\vec{r}_i - \vec{r}_j|} \, ,$$

(66)

where the second term accounts for the effective electric potential of the 3D QIT and the third term is responsible for the Coulomb repulsive force. In addition, we emphasize that the results obtained bring new contributions towards a better understanding of dynamical stability for charged particles levitated in a combined ion trap (Paul and Penning) [2], using both electrostatic DC and RF fields over which a constant static magnetic field is superimposed. Applications span areas of large interest such as stable confinement of antimatter and fundamental physics with antihydrogen [2,52]. We can also mention precision measurements and tests of the Standard Model using 3D QITs.

We can resume by stating that many body systems consisting of $N$ ions stored in a 3D QIT trap can be investigated locally in the neighbourhood of minimum configurations that characterize regular structures (Coulomb or ion crystals [46]). Collective models that

exhibit a small number of degrees of freedom can be introduced to achieve a comprehensive portrait of the system, or the electric potential can be estimated by means of particular potentials for which the $N$-particle potentials are integrable. Little perturbations generally preserve quantum stability. The many body system under investigation is also characterized by a continuous part of the energy spectrum, whose classical equivalent is achieved through a class of chaotic orbits. Nevertheless, a weak correspondence can be traced between classical and quantum nonlinear dynamics, based on Husimi functions [1,22]. As a result, it is straightforward to describe quantum ion crystals [53] by way of the minimum points associated to the Husimi function [37].

## 6. Highlights

We discuss dynamical stability for a classical system of two coupled oscillators in a 3D RF (Paul) trap using a well known model from literature [16–18], based on two control parameters: the axial angular moment and the ratio between the radial and axial secular frequencies of the trap. We enlarge the model by performing a qualitative analysis, based on the eigenvalues associated to the Hessian matrix of the potential, in order to explicitly determine the critical points, the minima and saddle points. The bifurcation set consists of the degenerate critical points. Its image in the control parameter space establishes the catastrophe set of equations which establishes the separatrix. We also supply the bifurcation diagram particularized to the system under investigation.

By illustrating the phase portraits we demonstrate that ion dynamics mainly consists of periodic and quasiperiodic trajectories, in the situation when the eigenfrequencies ratio is a rational number. In the scenario in which the eigenfrequencies ratio is an irrational number, the system is ergodic and it exhibits repetitive (iterative) rotations in the vicinity of a certain point. Our results also stand for ions with different masses or ions that exhibit different electrical charges, by generalizing the system investigated. By illustrating the phase portraits and the associated power spectra we show that ion dynamics is periodic or quasiperiodic for the parameter values employed in the numerical modeling.

The model we introduce is then used to investigate quantum stability for $N$ identical ions levitated in a 3D QIT, and we infer the elements of the Hessian matrix of the potential function $U$ in a critical point. We then apply our model to explore dynamical stability for SCCS consisting of $N$ identical ions confined in different types of 3D QIT (Paul, Penning, or combined traps) that exhibit cylindrical (axial) symmetry, and show they can be studied locally in the neighbourhood of the minimum configurations that describe ordered structures (Coulomb or ion crystals [46]). In order to perform a global description, we introduce collective models with a small number of degrees of freedom or the Coulomb potential can be approximated with specific potentials for which the $N$-particle potentials are integrable. Small enough perturbations maintain the quantum stability although the classical system may also exhibit a chaotic behavior.

We obtain the Hamilton function associated to a combined 3D QIT, which we show to be the sum of the Hamilton functions of the CM and of the relative motion of the ions. The ion distribution in the trap can be modeled by means of numerical analysis through the Hamilton function provided.

The results obtained bring new contributions towards a better understanding of the dynamical stability of charged particles in 3D QIT, and in particular in combined ion traps, with applications such as high precision mass spectrometry for elementary particles, search for spatio-temporal variations of the fundamental constants in physics at the cosmological scale, etc. Our approach is also very relevant in generalizing the parameters of different types of traps in a unified manner.

## 7. Conclusions

The paper suggests an alternative approach that is effective in describing the dynamical regimes for different types of traps in a coherent manner. The results obtained bring new contributions towards a better understanding of the dynamical stability (electrodynamics)

of charged particles in a combinational ion trap (Paul and Penning), using both electrostatic DC and RF fields over which a constant static magnetic field is superimposed. One of the advantages of our model lies in better characterizing ion dynamics for coupled two ion systems and for many body systems consisting of large number of ions. It also enables identifying stable solutions of motion and discussing the important issue of the critical points of the system, where the equilibrium configurations occur.

Applications span areas of vivid interest such as stable confinement of antimatter and fundamental physics with antihydrogen [2,52] or high precision measurements (including matter and antimatter tests of the Standard Model) [9,54]. Better characterization of ion dynamics in such traps would lead to longer trapping times, which is an issue of outmost importance. Other possible applications are Coulomb or ion crystals (multi body dynamics). The results and methods used are appropriate for the ion trap physics community to compare regimes without having the details of the trap itself.

**Author Contributions:** Conceptualization, B.M.M.; methodology, B.M.M. and S.L.; software, S.L.; validation, B.M.M. and S.L.; formal analysis, B.M.M.; investigation, B.M.M. and S.L.; resources, B.M.M. and S.L.; writing—original draft preparation, B.M.M.; writing—review and editing, B.M.M. and S.L.; visualization, S.L. and B.M.M.; project administration, B.M.M. All authors have read and agreed to the published version of the manuscript.

**Funding:** This research was funded by the Romanian Space Agency (ROSA)—contract number 136/2017 and by Ministerul Cercetării şi Inovării—contract number 16N/2019.

**Institutional Review Board Statement:** Not applicable.

**Informed Consent Statement:** Not applicable.

**Data Availability Statement:** The data that supports the results of this study are available within the article.

**Acknowledgments:** B.M.M. is grateful to Sabin Stoica for fruitful discussions and suggestions made before submitting the manuscript. B.M.M. and S.L. would like to thank the anonymous reviewers for the valuable suggestions and comments on the manuscript.

**Conflicts of Interest:** The authors declare no conflict of interest.

## Abbreviations

The following abbreviations are used in this manuscript:

| | |
|---|---|
| 3D | 3 Dimensional |
| CM | Centre of Mass |
| LIT | Linear Ion Trap |
| QED | Quantum Electrodynamics |
| QIP | Quantum Information Processing |
| QIT | Quadrupole Ion Trap |
| RF | Radiofrequency |
| SCCS | Strongly Coupled Coulomb Systems |
| SET | Surface Electrode Trap |

## Appendix A. Interaction Potential, Electric Potential of The Trap

We denote

$$\frac{k_1}{2} = Q_1 \beta_1, \tag{A1}$$

where $Q_1$ represents the electric charge of the ion labeled as 1. We assume the ions possess equal electric charges $Q_1 = Q_2$. The trap electric potential $\Phi_1 = \beta_1 x_1{}^2 + \dots$, can be considered as harmonic to a good approximation for the system of interest. In case of a

one-dimensional system of $s$ particles (ions) or for a system with $s$ degrees of freedom, the potential energy is:

$$U = \sum_{i=1}^{s} \frac{k_i \zeta_i^2}{2} + \frac{1}{2} \sum_{1 \leq i < j \leq s} b_{ij} \left( \zeta_i - \zeta_j \right)^2 , \tag{A2}$$

where $\zeta_i$ are the generalized coordinates and $\dot{\zeta}_i$ represent the generalized velocities. The electric potential is considered as a general solution of the Laplace equation, built using spherical harmonics functions with time dependent coefficients. By performing a series expansion of the Coulomb potential in spherical coordinates we can write down

$$\frac{1}{\left| \vec{x} - \vec{X} \right|} = \sum_{k=0}^{\infty} \left[ \begin{array}{c} r^k / R^{k+1} \ (\alpha) \\ R^k / r^{k+1} \ (\beta) \end{array} \right] \frac{4\pi}{2k+1} \sum_{q=-k}^{k} Y_{kq}^*(\Theta, \Phi) Y_{kq}(\theta, \varphi) , \tag{A3}$$

where $Y_{kq}^*$ and $Y_{kq}$ stand for the spherical harmonic functions. We choose $r = |\vec{x}|$ and $R = |\vec{X}|$. The expression labeled as $(\alpha)$ in Equation (A3) corresponds to the case $r < R$, while the expression labeled by $(\beta)$ is valid when $r > R$. We expand in series around $R$ assuming a diluted medium. We infer the interaction potential as

$$V_{int} = \frac{1}{4\pi\varepsilon_0} \sum_{1 \leq i < j \leq s} \frac{Q_i Q_j}{\left| \vec{r}_i - \vec{r}_j \right|} . \tag{A4}$$

## Appendix B. Dynamical Stability

As shown in Section 2.1, the expression of the autonomous Hamiltonian function associated to the system of two ions is given by Equation (24), where $r = \sqrt{\rho^2 + z^2}$, $\lambda = \mu_z / \mu_x$, $\mu_z = \sqrt{2(q^2 - a)}$. In fact $\lambda$ and $\nu$ represet the two control parameters chosen, with $\lambda$ the ratio between the secular axial and radial frequencies of the trap. $\nu$ denotes the scaled axial ($z$) component of the angular momentum $L_z$, while $\mu_z$ represents the second (or axial) secular frequency [16]. By calculus we infer

$$\lambda^2 = 4 \frac{q^2 - a}{q^2 + 2a}, \quad \nu^2 = \frac{2L_z^2}{q^2 + 2a} , \tag{A5}$$

and we discriminate among three cases [17]:

1. $\lambda = \frac{1}{2}$ and from Equation (A5) we infer

$$a = \frac{5q^2}{6}$$

2. $\lambda = 1$. Equation (A5) gives

$$a = \frac{q^2}{2}$$

3. $\lambda = 2$. By an analogous procedure we have

$$a = 0 .$$

## Appendix C. Quantum Stability

Using Equations (44) and (45) we obtain

$$\frac{\partial}{\partial q_{\gamma j}} \frac{1}{\left| \vec{r}_\alpha - \vec{r}_\beta \right|} = -\frac{1}{\left| \vec{r}_\alpha - \vec{r}_\beta \right|^2} \frac{\partial}{\partial q_{\gamma j}} \left| \vec{r}_\alpha - \vec{r}_\beta \right| \tag{A6}$$

We also have

$$|\vec{r}_\alpha - \vec{r}_\beta| = \sqrt{\sum_{h=1}^{3} \left( q_{\alpha h} - q_{\beta h} \right)^2} \; . \tag{A7}$$

Then

$$\frac{\partial}{\partial q_{\gamma j}} |\vec{r}_\alpha - \vec{r}_\beta| = |\vec{r}_\alpha - \vec{r}_\beta|^{-1} \left( q_{\alpha j} - q_{\beta j} \right) \left( \delta_{\alpha \gamma} - \delta_{\beta \gamma} \right), \tag{A8}$$

and

$$\frac{\partial}{\partial q_{\gamma j}} \frac{1}{|\vec{r}_\alpha - \vec{r}_\beta|} = -\frac{1}{|\vec{r}_\alpha - \vec{r}_\beta|^3} \left( q_{\alpha j} - q_{\beta j} \right) \left( \delta_{\alpha \gamma} - \delta_{\beta \gamma} \right). \tag{A9}$$

By using Equation (47) the last term in Equation (51) can be expressed as

$$\frac{\partial \zeta_{\alpha \gamma}}{\partial q_{\gamma' j'}} = \frac{Q_\alpha Q_\gamma}{4 \pi \varepsilon_0} \frac{\partial}{\partial q_{\gamma' j'}} \frac{1}{|\vec{r}_\alpha - \vec{r}_\gamma|^3} \; . \tag{A10}$$

Moreover

$$\frac{\partial}{\partial q_{\gamma' j'}} |\vec{r}_\alpha - \vec{r}_\gamma|^{-3} = -3 |\vec{r}_\alpha - \vec{r}_\gamma|^{-5} \left( q_{\alpha j'} - q_{\gamma j'} \right) \left( \delta_{\alpha \gamma'} - \delta_{\gamma \gamma'} \right). \tag{A11}$$

Then, Equation (A10) can be cast into

$$\frac{\partial \zeta_{\alpha \gamma}}{\partial q_{\gamma' j'}} = -\eta_{\alpha \gamma} \left( q_{\alpha j'} - q_{\gamma j'} \right) \left( \delta_{\alpha \gamma'} - \delta_{\gamma \gamma'} \right); \; \eta_{\alpha \gamma} = \frac{Q_\alpha Q_\gamma}{4 \pi \varepsilon_0} 3 |\vec{r}_\alpha - \vec{r}_\gamma|^{-5}, \; \alpha \neq \gamma \; . \tag{A12}$$

We use

$$\sum_{\alpha=1}^{N} q_\alpha \delta_{\alpha \gamma} = q_\gamma \; . \tag{A13}$$

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
