# Peer review of "Investigations on Dynamical Stability in 3D Quadrupole Ion Traps"

_applsci, doi:10.3390/app11072938_

Round 1

Reviewer 1 Report

The manuscript impresses by an uncommonly fluent English language, which is a pleasure to read. 
However, the manuscript is also affected by a common disease, the perceived need to mention a variety of high-flying projects and papers that are supposed to show the important stage on which the present project is set to also star. All those colourful clouds and sizzling fireworks - as if people did no longer do their own work in a much more basic reality. For my taste, most of the first about 28 citations are not really needed here, but they do not hurt either. 

The phase portraits (Figures 1-4) are extremely helpful in visualizing a number of the topics mentioned in the equational text. The central concept of trapping two different species simultaneously is certainly worth studying and optimizing. Concepts to this goal have been discussed since more than 20 years. 
The present manuscript is an elegant and highly readable contribution to the field, which I am happy to recommend for publication in ApplSci.  

Author Response

It is not our intention to impress or to be flamboyant when citing the bibliographical references used in the introductive part. We referred to key books such as Charged Particle Traps and Fundamental Physics in Particle Traps, which represent milestones in the domain. Our only aim was to emphasize the prominent role of ion traps in establishing novel experimental protocols and high precision techniques for quantum physics and quantum metrology, for atomic and nuclear physics. We have also cited previous papers published in the domain which we considered relevant, both as results and to point out that this paper extends our previous work.

We have implemented the suggestions and reduced the number of bibliographical references in the Introduction section by half. The section has been rewritten as we have tried to increase coherence. On the other hand, we have introduced 2 new references as another Reviewer has asked for extra references with respect to the scale of achievable trapping times in ion traps.

In the end we would like to extend our thanks to Reviewer # 1 for evaluating the manuscript, and for supplying useful comments and suggestions which have greatly improved the paper.

Reviewer 2 Report

This paper analyzes the stability when two or more ions are trapped simultaneously. Unfortunately the meaning is not clear.  Figures show the phase, but also the meaning of "phase" is not clear.

And I am not sure, if it is useful to discuss the stability using just the "pseudopotential".  This paper does not show the discussion with the real rf-trapping field. 

Therefore, I cannot recommend publication with the current form. 

At least, the clear explanation about the stability with a simple word should be required, particularly the motion of two ions have been already discussed, for example as shown in the reference Phys. Rev. A 40  808 (1989)

Author Response

We extend our thanks to Reviewer #2 for evaluating the manuscript and we consider that the comments and suggestions help us in better clarifying the significance of the results and their impact.

As indicated by the Reviewer, we have modified the introductory section, by augmenting its coherence and by eliminating unintended redundancy that existed in the previous version of the manuscript. We have modified L21 - L29 and L33 - L35 in page 1, and L41 - L50 in page 2.

1. The phase portraits shown in Figures 1 and 2 depict the dynamics of the system, where the solutions of the equations of motion are given by eqs. 18a and 18b (page 5). Such solutions are characteristic to a superposition of two oscillations, where C1 ÷ C4 represent constant coefficients and φ1 ÷ φ4 are the corresponding phases. The meaning of the “phases” is thus clearly described by eqs. 18a and 18b. The paper also supplies the modes of oscillation (the eigenfrequencies) under strong coupling and weak coupling conditions, where the latter case is inappropriate in practice.

2. The frequency of an atomic transition of an atom or single ion (quantum absorber) can be perturbed by many effects, among which we can enumerate the presence of parasitic electric and magnetic fields, an absorber that is not perfectly at rest, collisional effects or even imperfect vacuum conditions. The outcome of such systematic effects has been treated extensively in the literature, both experimentally and analytically. In case of optical clocks based on trapped ions we can also add quantum time dilation, an effect caused by the clock motion in a superposition of relativistic momentum wave packets.

The anharmonicity of the Coulomb interaction also induces large systematic effects, which cannot be disregarded in case of high precision measurements, such as optical clocks, ultrahigh precision spectroscopy, quantum logic and quantum metrology experiments, etc. These effects can be minimized due to the high sensitivity of the optical detection method when the ions are cooled to the ground state of motion, but this is far beyond the purpose of our paper.

As a matter of fact, the time-independent approximation that assumes the existence of a region around the effective trap minimum in which the potential is approximately a quadrupole, is valid for our system [as also shown in Phys. Rev A 99 043421 (2019)]. The simplest non-trivial model to describe the dynamic behaviour is the Hamilton function of the relative motion of two levitated ions that interact via the Coulomb force in an axial 3D QIT, under the pseudopotential approximation, as shown in Phys. Rev. A 40 808 (1989), Phys. Rev. A 48 3082 (1993) and Phys. Rev. A 49. The paper uses this well-established model, which we augment and extend.

Moreover, for values of the a-dimensional trap parameters a, q <<1 - such as is our case - the pseudopotential approximation holds very well.

Nevertheless, we would like to emphasize that we have investigated the effects induced by trap anharmonicities in a large number of papers recently published, which we exemplify below :

a. B. Mihalcea and G. Visan, Phys. Scr. T140 014057 (2010);

The paper explores ion (particle) dynamics of a Duffing oscillator confined in a nonlinear quadrupole Paul trap. In addition, we have considered that the particle undergoes interaction with a laser field in presence of a quartic potential (octupole anharmonicity). The phase portraits clearly reflect the existence of one or two attractors and of fractal basin boundaries for the trapped ion, assimilated with a periodically forced double-well oscillator. For particular initial conditions, some of the solutions obtained present a certain degree of periodicity, although the dynamics is irregular.

b. B. Mihalcea, Phys. Scr. T141 014018 (2011)

The paper investigates semiclassical dynamics for an ion confined in a nonlinear trap (higher order terms of the electric potential are considered). The electric potential exhibits both axial symmetry, as well as radial symmetry with respect to the xOy plan, which is characteristic for a large class of potentials that describes 3D Quadrupole Ion Traps (QIT) such as Paul, Penning and combined traps. We have shown that the quantum Hamiltonian describes an algebraic model when the anharmonic part is a polynomial function in the generators of the Lie algebra associated to the symplectic group Sp(2, R). This formalism can be applied to Hamiltonians that are nonlinear in the infinitesimal generators of a dynamical symmetry group, such as the case of ions confined in electrodynamic traps.

c. B. Mihalcea, Annals Phys. 388 (1) 100 (2018) and Rom. J. Phys. 62 (5 - 6) 113 (2017)

We propose a dequantization algorithm that enables explicit computation of the quantum Hamilton function associated to an anharmonic oscillator (ion) confined in a combined or RF trap, which describes an algebraic model when the anharmonic part is a polynomial. Such model is linear for any 3D QIT that exhibit axial symmetry. The method suggested in Phys. Scr. T141 is developed and particularized for both combined (Paul and Penning) as well as RF (Paul) traps, considering a RF anharmonic electric potential. Then, we build the classical Hamilton functions for such particular traps and find the classical Hamilton equations of motion for the anharmonic combined trap.

As emphasized above, in our previous work we have investigated both classical dynamics in nonlinear Paul traps, in presence of damping and laser radiation (a), and quantum dynamics of ions in axially symmetric quadrupole and octupole ion traps (b and c).

Nevertheless, we have added Appendix A to the paper, where we discuss the interaction potential and the electric potential for the system of interest. We have also introduced two new paragraphs in page 4 (L135 – L147), where we explain why the time-independent approximation can be safely used and why this family of potentials describes our system well. The pseudopotential approximation leads to erroneous results in case of laser cooling dynamics and in situations that are far from equilibrium.

3. To clarify one of the aspects raised by the referee with respect to “the clear explanation about the stability with a simple word should be required” we can summarize :

We extend the model established by Blumel, Moore and Farrelly [16]-[18] by analysing the eigenvalues of the Hessian matrix to better characterize dynamical stability and the critical points of the system. Our approach can be used to explicitly determine the critical points, the minima and saddle points, which brings a clear contribution towards a better understanding of the concept of dynamical stability for two ion and multi-body ion systems. Our method is valid both for identical ions and for different ion species. The approach we use is then extended to investigate quantum dynamics for many ion systems and then applied to the real case of a combined ion trap, for which the Hamilton function we supply can be used to illustrate numerically ion dynamics and the points of minimum, where equilibrium configurations occur.

Reviewer 3 Report

The manuscript by B. M. Mihalcea and S. Lynch, starting from a well-known model for studying the dynamical stability of charged particles trapped in RF traps, reports on the extension of this model based on the use of the Hessian matrix of the potential function for better describing the dynamical stability and the critical points of ion trapped systems. The last part of the manuscript presents a focus on the characterization of the combined RF and Penning ion traps.

In general, the manuscript is well referenced and contains enough details for understanding and analyzing the presented work, therefore it is worth the publication. Nevertheless, some parts of the manuscript should be rephrased before the publication since the concepts are not always well presented. The abstract, for example, is substantially an outline resume and does not well highlight the lack of knowledge the authors aim at filling with this work.

The central sections of the manuscript are not well linked together and, through cases and subcases, it is easy to lose the track of the point. I would add some explanations about what the authors are going to show or what they just presented, basically moving the bullets from Sec. 6 to the place where they belong (instead of gather all together in another section). With this rearrangement of the text, Sec. 6 should be suppressed, since, again, does not present any result but a resume of the manuscript.

With the first part of Sec. 7, the manuscript seems quite repetitive and verbose since some concepts are reported for the third time to the reader attention. I suggest to merge the remaining parts of Sec. 6 and the first part of Sec. 7 in a sort of “recap” section (very useful to get the meaning of this work without going through all the mathematical details), and create a “conclusion” section (which in this version starts only from L. 437) where the authors emphasize their contributions and highlight the possible applications of their model.

In this regard, it would be very useful if the authors add a brief operative guideline about how to apply the model to a real experimental case.  

Other remarks:

  • L. 34 May the authors reference the statement about the ion trapping lifetime on the order of “months” and “years”?
  • L. 123 Sec. 3 is not presented in the paper outline.
  • L. 157, eq. 14, “-a” should be “-b”.
  • Graphs: the axes’ labels are tiny, thus unreadable.
  • Graphs: in general, to improve the readability of the graphs and for the sake of clearness, I suggest to report the initial conditions or the parameters which characterize each plot directly under the interested graph (e.g. “a)… ”).
  • Graphs: the captions of Figs. 3 and 4 are not clear and should be possibly rephrased (see previous bullet). In addition, there’s no description of graph d).

Author Response

Comments and Suggestions for Authors

The manuscript by B. M. Mihalcea and S. Lynch, starting from a well-known model for studying the dynamical stability of charged particles trapped in RF traps, reports on the extension of this model based on the use of the Hessian matrix of the potential function for better describing the dynamical stability points and the critical of ion trapped systems. The last part of the manuscript presents a focus on the characterization of the combined RF and Penning ion traps.

In general, the manuscript is well referenced and contains enough details for understanding and analyzing the presented work, therefore it is worth the publication. Nevertheless, some parts of the manuscript should be rephrased before the publication since the concepts are not always well presented. The abstract, for example, is substantially an outline resume and does not well highlight the lack of knowledge the authors aim at filling with this work.

Answer: We have rewritten the abstract according to the guidelines above, in an attempt to better emphasize the original contributions that the paper brings.

The central sections of the manuscript are not well linked together and, through cases and subcases, it is easy to lose the track of the point. I would add some explanations about what the authors are going to show or what they just presented, basically moving the bullets from Sec. 6 to the place where they belong (instead of gather all together in another section). With this rearrangement of the text, Sec. 6 should be suppressed, since, again, does not present any result but a resume of the manuscript.

With the first part of Sec. 7, the manuscript seems quite repetitive and verbose since some concepts are reported for the third time to the reader attention. I suggest to merge the remaining parts of Sec. 6 and the first part of Sec. 7 in a sort of “recap” section (very useful to get the meaning of this work without going through all the mathematical details), and create a “conclusion” section (which in this version starts only from L. 437) where the authors emphasize their contributions and highlight the possible applications of their model.

In this regard, it would be very useful if the authors add a brief operative guideline about how to apply the model to a real experimental case.

Answer: We have performed the changes as suggested, Sec. 6 was suppressed and replaced with Section Highlights. It is followed by the Conclusion Section.

With respect to the cases and subcases issue, we tried to make things as clear as it was possible, considering the mathematical apparatus we use. We refer to the two subsections, namely Subsection 3.2. Solutions of the equations of motion and Subsection 3.3. Critical Points. Discussion, that are clearly separated. Each one of these subsections consists only of two cases and a corresponding number of subcases, where we characterize the critical points (minimum, saddle point, degenerate critical point -as is the case for Subsection 3.3). The terminology is characteristic to the dynamical systems theory (Morse theory). We sincerely hope that “navigating” through this “menu” is not so complicated, but we could not find an even simpler way to do it.

We have included an operative guideline at the end of each Section, with an aim to explain the principle of the technique(s) on which our method is based. Thus, in Sec. 2 we supply the eigenfrequencies and the stable solution of the equation of motion, which enables the reader to represent phase portraits or power spectra for any range of parameters that one might choose.

The bifurcation diagram in Sec. 3 can also be easily reproduced, as the bifurcation set is supplied.

In addition, the Hamilton function that we supply at the end of Section 5 can be used to perform a numerical modeling of the system of many-body ions confined in any type of 2D or 3D QIT, and thus obtain the configurations of equilibrium and the ion distribution in space (2D or 3D).

Other remarks:

L. 34 May the authors reference the statement about the ion trapping lifetime on the order of “months” and “years”?

Answer: Bibliographical references are introduced that support and clear this aspect

L. 123 Sec. 3 is not presented in the paper outline.

Answer: Sec. 3 is now included in the paper outline

L. 157, eq. 14, “-a” should be “-b”.

Answer: the change was performed

Graphs: the axes’ labels are tiny, thus unreadable.

Graphs: in general, to improve the readability of the graphs and for the sake of clearness, I suggest to report the initial conditions or the parameters which characterize each plot directly under the

interested graph (e.g. “a)... ”).

Graphs: the captions of Figs. 3 and 4 are not clear and should be possibly rephrased (see previous bullet). In addition, there’s no description of graph d).

Answer: All the changes indicated have been performed in the updated version of the manuscript, all figures have been redrawn and the figure captions have been rephrased as suggested.

The authors would like to thank the anonymous reviewer for the attention paid to the manuscript, as well as for the valuable suggestions and comments that helped in clarifying several aspects of interest, while rendering the manuscript more coherent and easier to understand.

Round 2

Reviewer 2 Report

I think, the manuscript became easier to read, and I agree to publish it.